# A Stochastic Proximal Polyak Step Size

**Fabian Schaipp**                                                      *fabian.schaipp@tum.de*
*Department of Mathematics*
*Technical University of Munich*

**Robert M. Gower**                                              *rgower@flatironinstitute.org*
*Center for Computational Mathematics*
*Flatiron Institute, New York*

**Michael Ulbrich**                                                        *m.ulbrich@tum.de*
*Department of Mathematics*
*Technical University of Munich*

**Reviewed on OpenReview:** *https://openreview.net/forum?id=jWr41htaB3*

## Abstract

Recently, the stochastic Polyak step size (`SPS`) has emerged as a competitive adaptive step size scheme for stochastic gradient descent. Here we develop `ProxSPS`, a *proximal* variant of `SPS` that can handle regularization terms. Developing a proximal variant of `SPS` is particularly important, since `SPS` requires a lower bound of the objective function to work well. When the objective function is the sum of a loss and a regularizer, available estimates of a lower bound of the sum can be loose. In contrast, `ProxSPS` only requires a lower bound for the loss which is often readily available. As a consequence, we show that `ProxSPS` is easier to tune and more stable in the presence of regularization. Furthermore for image classification tasks, `ProxSPS` performs as well as `AdamW` with little to no tuning, and results in a network with smaller weight parameters. We also provide an extensive convergence analysis for `ProxSPS` that includes the non-smooth, smooth, weakly convex and strongly convex setting.

## 1 Introduction

Consider problems of the form

$$\min_{x \in \mathbb{R}^n} f(x), \quad f(x) := \mathbb{E}_P[f(x; S)] = \int_{\mathcal{S}} f(x; s) dP(s), \tag{1}$$

where $\mathcal{S}$ is a sample space (or sample set). Formally, we can see $S$ as a random variable mapping to $\mathcal{S}$ and $P(s)$ as the associated probability measure. Let us assume that for each $s \in \mathcal{S}$, the function $f(\cdot; s) : \mathbb{R}^n \to \mathbb{R}$ is locally Lipschitz and hence possesses the Clarke subdifferential $\partial f(\cdot; s)$ (Clarke, 1983). Problems of form (1) arise in machine learning tasks where $\mathcal{S}$ is the space of available data points (Bottou et al., 2018). An efficient method for such problems is stochastic (sub)gradient descent (Robbins & Monro, 1951; Bottou, 2010; Davis & Drusvyatskiy, 2019), given by

$$x^{k+1} = x^k - \alpha_k g_k, \quad g_k \in \partial f(x^k; S_k), \quad \text{where } S_k \sim P. \tag{SGD}$$

Moreover, we will also consider the composite problem

$$\min_{x \in \mathbb{R}^n} \psi(x), \quad \psi(x) := f(x) + \varphi(x), \tag{2}$$

where $\varphi : \mathbb{R}^n \to \mathbb{R} \cup \{\infty\}$ is a proper, closed, and convex regularization function. In practical situations, the expectation in the objective function $f$ is typically approximated by a sample average over $N \in \mathbb{N}$ data points. We formalize this special case with

$$\mathcal{S} = \{s_1, \ldots, s_N\}, \ P(s_i) = \frac{1}{N}, \ f_i := f(\cdot; s_i) \quad i = 1, \ldots, N. \tag{ER}$$

In this case, problem (1) becomes the empirical risk minimization problem

$$\min_{x \in \mathbb{R}^n} \frac{1}{N} \sum_{i=1}^{N} f_i(x).$$

## 1.1 Background and Contributions

**Polyak step size.** For minimizing a convex, possibly non-differentiable function $f$, Polyak (1987, Chapter 5.3) proposed

$$x^{k+1} = x^k - \alpha_k g_k, \quad \alpha_k = \frac{f(x^k) - \min f}{\|g_k\|^2}, \quad g_k \in \partial f(x^k) \setminus \{0\}.$$

This particular choice of $\alpha_k$, requiring the knowledge of $\min f$, has been subsequently called the *Polyak step size* for the subgradient method. Recently, Berrada et al. (2019); Loizou et al. (2021); Orvieto et al. (2022) adapted the Polyak step size to the stochastic setting: consider the (ER) case and assume that each $f_i$ is differentiable and that a lower bound $C(s_i) \leq \inf_x f_i(x)$ is known for all $i \in [N]$. The method proposed by (Loizou et al., 2021) is

$$x^{k+1} = x^k - \min\left\{\gamma_b, \frac{f_{i_k}(x^k) - C(s_{i_k})}{c\|\nabla f_{i_k}(x^k)\|^2}\right\}\nabla f_{i_k}(x^k), \tag{$\text{SPS}_{\max}$}$$

with hyper-parameters $c, \gamma_b > 0$ and where in each iteration $i_k$ is drawn from $\{1, \ldots, N\}$ uniformly at random. It is important to note that the initial work (Loizou et al., 2021) used $C(s_i) = \inf f_i$; later, Orvieto et al. (2022) established theory for ($\text{SPS}_{\max}$) for the more general case of $C(s_i) \leq \inf_x f_i(x)$ and allowing for mini-batching. Other works analyzed the Polyak step size in the convex, smooth setting (Hazan & Kakade, 2019) and in the convex, smooth and stochastic setting (Prazeres & Oberman, 2021). Further, the stochastic Polyak step size is closely related to stochastic model-based proximal point (Asi & Duchi, 2019) as well as stochastic bundle methods (Paren et al., 2022).

*Contribution.* We propose a proximal version of the stochastic Polyak step size, called `ProxSPS`, which explicitly handles regularization functions. Our proposal is based crucially on the fact that the stochastic Polyak step size can be motivated with stochastic proximal point for a truncated linear model of the objective function (we explain this in detail in Section 3.1). Our method has closed-form updates for squared $\ell_2$-regularization. We provide theoretical guarantees for `ProxSPS` for any closed, proper, and convex regularization function (including indicator functions for constraints). Our main results, Theorem 7 and Theorem 8, also give new insights for $\text{SPS}_{\max}$, in particular showing exact convergence for convex and non-convex settings.

**Lower bounds and regularization.** Methods such as $\text{SPS}_{\max}$ need to estimate a lower bound $C(s)$ for each loss function $f(\cdot; s)$. Though $\inf_x f(x; s)$ can be precomputed in some restricted settings, in practice the lower bound $C(s) = 0$ is used for non-negative loss functions.[1] The tightness of the choice $C(s)$ is further reflected in the constant $\sigma^2 := \min f - \mathbb{E}_P[C(S)]$, which affects the convergence guarantees of $\text{SPS}_{\max}$ (Orvieto et al., 2022).

*Contribution.* For regularized problems (2) and if $\varphi$ is differentiable, the current proposal of $\text{SPS}_{\max}$ would add $\varphi$ to every loss function $f(\cdot; s)$. In this case, for non-negative regularization terms, such as the squared $\ell_2$-norm, the lower bound $C(s) = 0$ is always loose. Indeed, if $\varphi \geq 0$, then $\inf_{x \in \mathbb{R}^n}(f(x; s) + \varphi(x)) \geq \inf_{x \in \mathbb{R}^n} f(x; s)$ and this inequality is strict in most practical scenarios. For our proposed method `ProxSPS`,

---

[1]See for instance https://github.com/IssamLaradji/sps.

we now need only estimate a lower bound for the loss $f(x; s)$ and not for the composite function $f(x; s)+\varphi(x)$. Further, `ProxSPS` decouples the adaptive step size for the gradient of the loss from the regularization (we explain this in detail in Section 4.1 and Fig. 1).

**Proximal and adaptive methods.** The question on how to handle regularization terms has also been posed for other families of adaptive methods. For `Adam` (Kingma & Ba, 2015) with $\ell_2$-regularization it has been observed that it generalizes worse and is harder to tune than `AdamW` (Loshchilov & Hutter, 2019) which uses weight decay. Further, `AdamW` can be seen as an approximation to a proximal version of `Adam` (Zhuang et al., 2022).[2] On the other hand, Loizou et al. (2021) showed that – without regularization – default hyperparameter settings for $\text{SPS}_{\max}$ give very encouraging results on matrix factorization and image classification tasks. This is promising since it suggests that $\text{SPS}_{\max}$ is an *adaptive* method, and can work well across varied tasks without the need for extensive hyperparameter tuning.

*Contribution.* We show that by handling $\ell_2$-regularization using a proximal step, our resulting `ProxSPS` is less sensitive to hyperparameter choice as compared to $\text{SPS}_{\max}$. This becomes apparent in matrix factorization problems, where `ProxSPS` converges for a much wider range of regularization parameters and learning rates, while $\text{SPS}_{\max}$ is more sensitive to these settings. We also show similar results for image classification over the `CIFAR10` and `Imagenet32` dataset when using a `ResNet` model, where, compared to `AdamW`, our method is less sensitive with respect to the regularization parameter.

The remainder of our paper is organized as follows: we will first recall how the stochastic Polyak step size, in the case of $\varphi = 0$, can be derived using the model-based approach of (Asi & Duchi, 2019; Davis & Drusvyatskiy, 2019) and how this is connected to $\text{SPS}_{\max}$. We then derive `ProxSPS` based on the connection to model-based methods, and present our theoretical results, based on the proof techniques in (Davis & Drusvyatskiy, 2019).

## 2 Preliminaries

### 2.1 Notation

Throughout, we will write $\mathbb{E}$ instead of $\mathbb{E}_P$. For any random variable $X(s)$, we denote $\mathbb{E}[X(S)] := \int_{\mathcal{S}} X(s)dP(s)$. We denote $(\cdot)_+ := \max\{\cdot, 0\}$. We write $\tilde{\mathcal{O}}$ when we drop logarithmic terms in the $\mathcal{O}$-notation, e.g. $\tilde{\mathcal{O}}(\frac{1}{K}) = \mathcal{O}(\frac{\ln(1+K)}{K})$.

### 2.2 General assumptions

Throughout the article, we assume the following:

**Assumption 1.** *It is possible to generate infinitely many i.i.d. realizations $S_1, S_2, \ldots$ from $\mathcal{S}$.*

**Assumption 2.** *For every $s \in \mathcal{S}$, $\inf_x f(x; s)$ is finite and there exists $C(s)$ satisfying $C(s) \leq \inf_x f(x; s)$.*

In many machine learning applications, non-negative loss functions are used and thus we can satisfy the second assumption choosing $C(s) = 0$ for all $s \in \mathcal{S}$.

### 2.3 Convex analysis

Let $h : \mathbb{R}^n \to \mathbb{R}$ be convex and $\alpha > 0$. The proximal operator is given by

$$\text{prox}_{\alpha h}(x) := \arg\min_y h(y) + \frac{1}{2\alpha}\|y - x\|^2.$$

Further, the Moreau envelope is defined by $\text{env}_h^\alpha(x) := \min_y h(y) + \frac{1}{2\alpha}\|y - x\|^2$, and its derivative is $\nabla\text{env}_h^\alpha(x) = \frac{1}{\alpha}(x - \text{prox}_{\alpha h}(x))$ (Drusvyatskiy & Paquette, 2019, Lem. 2.1). Moreover, due to the optimality conditions of the proximal operator, if $h \in \mathcal{C}^1$ then

$$\hat{x} = \text{prox}_{\alpha h}(x) \implies \|\nabla h(\hat{x})\| = \alpha^{-1}\|x - \hat{x}\| = \|\nabla\text{env}_h^\alpha(x)\|. \tag{3}$$

---

[2]For `SGD` treating $\ell_2$-regularization as a part of the loss can be seen to be equivalent to its proximal version (cf. Appendix C).

Davis & Drusvyatskiy (2019) showed how to use the Moreau envelope as a measure of stationarity: if $\|\nabla \mathrm{env}_h^\alpha(x)\|$ is small, then $x$ is close to $\hat{x}$ and $\hat{x}$ is an almost stationary point of $h$. Formally, the gradient of the Moreau envelope can be related to the gradient mapping (cf. (Drusvyatskiy & Paquette, 2019, Thm. 4.5) and Lemma 11).

We say that a function $h : \mathbb{R}^n \to \mathbb{R}$ is $L$-smooth if its gradient is $L$–Lipschitz, that is

$$\|\nabla h(x) - \nabla h(y)\| \ \leq \ L \|x - y\|, \quad \forall x, y \in \mathbb{R}^n. \tag{4}$$

If $h$ is $L$-smooth, then

$$h(y) \leq h(x) + \langle \nabla h(x), y - x \rangle + \frac{L}{2} \|y - x\|^2 \quad \text{for all } x, y, \in \mathbb{R}^n.$$

A function $h : \mathbb{R}^n \to \mathbb{R}$ is $\rho$–weakly convex for $\rho \geq 0$ if $h + \frac{\rho}{2} \|\cdot\|^2$ is convex. Any $L$–smooth function is weakly convex with parameter less than or equal to $L$ (Drusvyatskiy & Paquette, 2019, Lem. 4.2). The above results on the proximal operator and Moreau envelope can immediately be extended to $h$ being $\rho$–weakly convex if $\alpha \in (0, \rho^{-1})$, since then $h + \frac{\rho}{2} \|\cdot\|^2$ is convex.

If we assume that each $f(\cdot; s)$ is $\rho_s$-weakly convex for $\rho_s \geq 0$, then applying (Bertsekas, 1973, Lem. 2.1) to the convex function $f(\cdot; s) + \frac{\rho_s}{2} \|\cdot\|^2$ yields that $f + \frac{\rho}{2} \|\cdot\|^2$ is convex and thus $f$ is $\rho$–weakly convex for $\rho := \mathbb{E}[\rho_S]$. In particular, $f$ is convex if each $f(\cdot; s)$ is assumed to be convex. For a weakly convex function $h$, we denote with $\partial h$ the regular subdifferential (cf. (Davis & Drusvyatskiy, 2019, section 2.2) and (Rockafellar & Wets, 1998, Def. 8.3)).

## 3 The unregularized case

For this section, consider problems of form (1), i.e. no regularization term $\varphi$ is added to the loss $f$.

### 3.1 A model-based view point

Many classical methods for solving (1) can be summarized by model-based stochastic proximal point: in each iteration, a model $f_x(\cdot; s)$ is constructed approximating $f(\cdot; s)$ locally around $x$. With $S_k \sim P$ being drawn at random, this yields the update

$$x^{k+1} = \arg\min_{y \in \mathbb{R}^n} f_{x^k}(y; S_k) + \frac{1}{2\alpha_k} \|y - x^k\|^2. \tag{5}$$

The theoretical foundation for this family of methods has been established by Asi & Duchi (2019) and Davis & Drusvyatskiy (2019). They give the following three models as examples:

(i) *Linear:* $f_x(y; s) := f(x; s) + \langle g, y - x \rangle$ with $g \in \partial f(x; s)$.

(ii) *Full:* $f_x(y; s) := f(y; s)$.

(iii) *Truncated:* $f_x(y; s) := \max\{f(x; s) + \langle g, y - x \rangle, \inf_{z \in \mathbb{R}^n} f(z; s)\}$ where $g \in \partial f(x; s)$.

It is easy to see that update (5) for the *linear model* is equal to (SGD) while the *full model* results in the stochastic proximal point method. For the *truncated model*, (5) results in the update

$$x^{k+1} = x^k - \min\left\{\alpha_k, \frac{f(x^k; S_k) - \inf_{z \in \mathbb{R}^n} f(z; S_k)}{\|g_k\|^2}\right\} g_k, \quad g_k \in \partial f(x^k, S_k). \tag{6}$$

More generally, one can replace the term $\inf_{x \in \mathbb{R}^n} f(x; S_k)$ with an arbitrary lower bound of $f(\cdot; S_k)$ (cf. Lemma 10). The model-based stochastic proximal point method for the truncated model is given in Algorithm 1. The connection between the truncated model and the method depicted in (6) is not a new insight and has been pointed out in several works (including (Asi & Duchi, 2019; Loizou et al., 2021) and (Berrada

et al., 2019, Prop. 1)). For simplicity, we refer to Algorithm 1 as `SPS` throughout this article. However, it should be pointed out that this acronym (and variations of it) have been used for stochastic Polyak-type methods in slightly different ways (Loizou et al., 2021; Gower et al., 2021).

---

**Algorithm 1** `SPS`

---

**Require:** $x^0 \in \mathbb{R}^n$, step sizes $\alpha_k > 0$.
    **for** $k = 0, 1, 2, \ldots, K - 1$ **do**
        1. Sample $S_k$ and set $C_k := C(S_k)$.
        2. Choose $g_k \in \partial f(x^k; S_k)$. If $g_k = 0$, set $x^{k+1} = x^k$. Otherwise, set

$$x^{k+1} = x^k - \gamma_k g_k, \quad \gamma_k = \min\left\{\alpha_k, \frac{f(x^k; S_k) - C_k}{\|g_k\|^2}\right\}. \tag{7}$$

    **return** $x^K$

---

For instance consider again the $\text{SPS}_{\max}$ method

$$x^{k+1} = x^k - \min\left\{\gamma_b, \frac{f_{i_k}(x^k) - C(s_{i_k})}{c\|\nabla f_{i_k}(x^k)\|^2}\right\}\nabla f_{i_k}(x^k), \tag{$\text{SPS}_{\max}$}$$

where $c, \gamma_b > 0$. Clearly, for $c = 1$ and $\alpha_k = \gamma_b$, update (7) is identical to $\text{SPS}_{\max}$. With this in mind, we can interpret the hyperparameter $\gamma_b$ in $\text{SPS}_{\max}$ simply as a step size for the model-based stochastic proximal point step. For the parameter $c$ on the other hand, the model-based approach motivates the choice $c = 1$. In this article, we will focus on this natural choice $c = 1$ which also reduces the amount of hyperparameter tuning. However, we should point out that, in the strongly convex case, $c = 1/2$ gives the best rate of convergence in (Loizou et al., 2021).

## 4 The regularized case

Now we consider regularized problems of the form (2), i.e.

$$\min_{x \in \mathbb{R}^n} \psi(x), \quad \psi(x) = f(x) + \varphi(x),$$

where $\varphi : \mathbb{R}^n \to \mathbb{R} \cup \{\infty\}$ is a proper, closed, $\lambda$-strongly convex function for $\lambda \geq 0$ (we allow $\lambda = 0$). For $s \in \mathcal{S}$, denote by $\psi_x(\cdot; s)$ a stochastic model of the objective $\psi$ at $x$. We aim to analyze algorithms with the update

$$x^{k+1} = \arg\min_{x \in \mathbb{R}^n} \psi_{x^k}(x; S_k) + \frac{1}{2\alpha_k}\|x - x^k\|^2, \tag{8}$$

where $S_k \sim P$ and $\alpha_k > 0$. Naively, if we know a lower bound $\tilde{C}(s)$ of $f(\cdot; s) + \varphi(\cdot)$, the truncated model could be constructed for the function $f(x; s) + \varphi(x)$, resulting in

$$\psi_x(y; s) = \max\{f(x; s) + \varphi(x) + \langle g + u, y - x\rangle, \tilde{C}(s)\}, \quad g \in \partial f(x; s), \quad u \in \partial \varphi(x). \tag{9}$$

In fact, Asi & Duchi (2019) and Loizou et al. (2021) work in the setting of unregularized problems and hence their approaches would handle regularization in this way. What we propose instead, is to only truncate a linearization of the loss $f(x; s)$, yielding the model

$$\psi_x(y; s) = f_x(y; s) + \varphi(y), \quad f_x(y; s) = \max\{f(x; s) + \langle g, y - x\rangle, C(s)\}, \quad g \in \partial f(x; s). \tag{10}$$

Solving (8) with the model in (10) results in

$$x^{k+1} = \arg\min_{y \in \mathbb{R}^n} \max\{f(x^k; S_k) + \langle g_k, y - x^k\rangle, C(S_k)\} + \varphi(y) + \frac{1}{2\alpha_k}\|y - x^k\|^2. \tag{11}$$

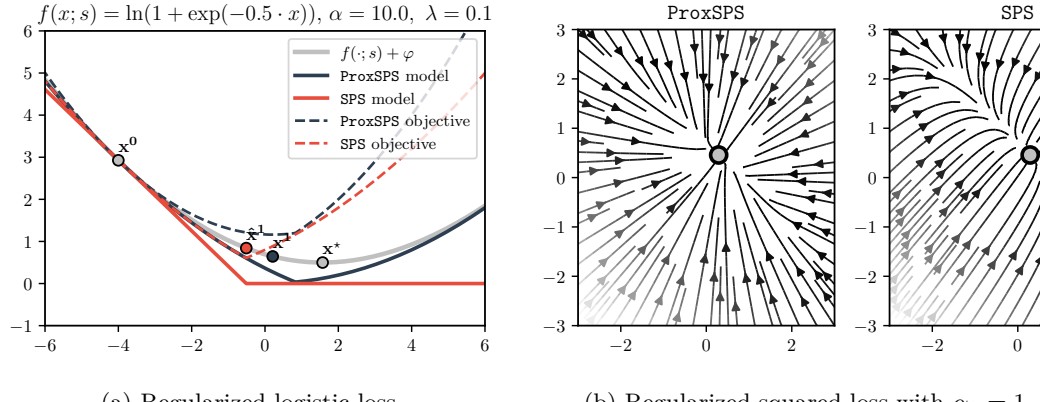

(a) Regularized logistic loss.

(b) Regularized squared loss with $\alpha_k = 1$, $\lambda = 1$.

Figure 1: a) `SPS` refers to model (9) whereas `ProxSPS` refers to (10). We plot the corresponding model $\psi_{x^0}(y; s)$ and the objective function of (8). $x^1$ (resp. $\hat{x}^1$) denotes the new iterate for `ProxSPS` (resp. `SPS`), $x^\star$ is the minimizer of $f(\cdot; s) + \varphi$. b) Streamlines of the vector field $V(x^k) := x^{k+1} - x^k$, for $f(x) = \|Ax - b\|^2$ and for the deterministic update, i.e. $f(x; s) = f(x)$. `ProxSPS` refers to update (14) and `SPS` refers to (13). The circle marks the minimizer of $f(x) + \frac{\lambda}{2}\|x\|^2$.

The resulting model-based stochastic proximal point method is given in Algorithm 2 [3]. Lemma 12 shows that, if $\mathrm{prox}_\varphi$ is known, update (11) can be computed by minimizing a strongly convex function over a compact one-dimensional interval. The relation to the proximal operator of $\varphi$ motivates the name `ProxSPS`. Further, the `ProxSPS` update (11) has a closed form solution when $\varphi$ is the squared $\ell_2$-norm, as we detail in the next section.

---

**Algorithm 2** `ProxSPS`

---

**Require:** $x^0 \in \mathbb{R}^n$, step sizes $\alpha_k > 0$.
    **for** $k = 0, 1, 2, \ldots, K - 1$ **do**
        1. Sample $S_k$ and set $C_k := C(S_k)$.
        2. Choose $g_k \in \partial f(x^k; S_k)$.
        Update $x^{k+1}$ according to (11).
    **return** $x^K$

---

### 4.1 The special case of $\ell_2$-regularization

When $\varphi(x) = \frac{\lambda}{2}\|x\|^2$ for some $\lambda > 0$, `ProxSPS` (11) has a closed form solution as we show next in Lemma 1. For this lemma, recall that the proximal operator of $\varphi(x) = \frac{\lambda}{2}\|x\|^2$ is given by $\mathrm{prox}_{\alpha\varphi}(x) = \frac{1}{1+\alpha\lambda}x$ for all $\alpha > 0$, $x \in \mathbb{R}^n$.

**Lemma 1.** *Let $\varphi(x) = \frac{\lambda}{2}\|x\|^2$ and let $g \in \partial f(x; s)$ and $C(s) \leq \inf_{z \in \mathbb{R}^n} f(z; s)$ hold for all $s \in \mathcal{S}$. For $\psi_x(y; s) = f_x(y; s) + \varphi(y)$ with $f_x(y; s) = \max\{f(x; s) + \langle g, y - x \rangle, C(s)\}$ consider the update*

$$x^{k+1} = \arg\min_{x \in \mathbb{R}^n} \psi_{x^k}(x; S_k) + \frac{1}{2\alpha_k}\|x - x^k\|^2.$$

*Denote $C_k := C(S_k)$ and let $g_k \in \partial f(x^k; S_k)$. Define*

$$\tau_k^+ := \begin{cases} 0 & \text{if } g_k = 0, \\ \min\left\{\alpha_k, \left(\frac{(1+\alpha_k\lambda)(f(x^k; S_k) - C_k) - \alpha_k\lambda\langle g_k, x^k\rangle}{\|g_k\|^2}\right)_+\right\} & \text{else.} \end{cases}$$

---

[3]For $\varphi = 0$, Algorithm 2 is identical to Algorithm 1.

*Update* (11) *is given by*

$$x^{k+1} \;=\; \frac{1}{1+\alpha_k\lambda}\Big(x^k - \tau_k^+ g_k\Big) = \text{prox}_{\alpha_k\varphi}(x^k - \tau_k^+ g_k). \tag{12}$$

See Lemma 9 in the appendix for an extended version of the above lemma and its proof. The update (12) can be naturally decomposed into two steps, one stochastic gradient step with an adaptive stepsize, that is $\bar{x}^{k+1} = x^k - \tau_k^+ g_k$ followed by a proximal step $x^{k+1} = \text{prox}_{\alpha_k\varphi}(\bar{x}^{k+1})$. This decoupling into two steps, makes it easier to interpret the effect of each step, with $\tau_k^+$ adjusting for the scale/curvature and the following proximal step shrinking the resulting parameters. There is no clear separation of tasks if we apply the `SPS` method to the regularized problem, as we see next.

---

**Algorithm 3** `ProxSPS` for $\varphi = \frac{\lambda}{2}\|\cdot\|^2$

---

**Require:** $x^0 \in \mathbb{R}^n$, step sizes $\alpha_k > 0$.
    **for** $k = 0, 1, 2, \ldots, K-1$ **do**
       1. Sample $S_k$ and set $C_k := C(S_k)$.
       2. Choose $g_k \in \partial f(x^k; S_k)$. If $g_k = 0$, set $x^{k+1} = \frac{1}{1+\alpha_k\lambda}x^k$. Otherwise, set

$$x^{k+1} = \frac{1}{1+\alpha_k\lambda}\left[x^k - \min\left\{\alpha_k, \left(\frac{(1+\alpha_k\lambda)(f(x^k;S_k) - C_k) - \alpha_k\lambda\langle g_k, x^k\rangle}{\|g_k\|^2}\right)_+\right\}g_k\right].$$

    **return** $x^K$

---

### 4.2 Comparing the model of `SPS` and `ProxSPS`

For simplicity, assume again the discrete sample space setting (ER) with differentiable loss functions $f_i$ and let $\varphi = \frac{\lambda}{2}\|\cdot\|^2$. Clearly, the composite problem (2) can be transformed to an instance of (1) by setting $\ell_i(x) := f_i(x) + \frac{\lambda}{2}\|x\|^2$ and solving $\min_x \ell(x)$ with $\ell(x) := \frac{1}{N}\sum_{i=1}^{N}\ell_i(x)$. Assume that a lower bound $\underline{\ell}_i \leq \inf_x \ell_i(x)$ is known. In this case (9) becomes

$$\psi_x(y; s_i) = \max\left\{f_i(x) + \tfrac{\lambda}{2}\|x\|^2 + \langle \nabla f_i(x) + \lambda x, y - x\rangle,\; \underline{\ell}_i\right\}.$$

Due to Lemma 10, if $\nabla f_{i_k}(x^k) + \lambda x^k \neq 0$, the update (8) is given by

$$x^{k+1} = x^k - \min\left\{\alpha_k, \frac{f_{i_k}(x^k) + \frac{\lambda}{2}\|x^k\|^2 - \underline{\ell}_{i_k}}{\|\nabla f_{i_k}(x^k) + \lambda x^k\|^2}\right\}(\nabla f_{i_k}(x^k) + \lambda x^k). \tag{13}$$

We refer to this method, which is using model (9), as `SPS`. On the other hand, using model (10) and if $\nabla f_{i_k}(x^k) \neq 0$, the update of `ProxSPS` (12) is

$$x^{k+1} = \frac{1}{1+\alpha_k\lambda}\left[x^k - \min\left\{\alpha_k, \left(\frac{(1+\alpha_k\lambda)(f_{i_k}(x^k) - C(s_{i_k})) - \alpha_k\lambda\langle\nabla f_{i_k}(x^k), x^k\rangle}{\|\nabla f_{i_k}(x^k)\|^2}\right)_+\right\}\nabla f_{i_k}(x^k)\right]. \tag{14}$$

In Fig. 1a, we illustrate the two models (9) (denoted by `SPS`) and (10) (denoted by `ProxSPS`) for the logistic loss with squared $\ell_2$-regularization. We can see that the `ProxSPS` model is a much better approximation of the (stochastic) objective function as it still captures the quadratic behaviour of $\varphi$. Furthermore, as noted in the previous section, `ProxSPS` decouples the step size of the gradient and of the shrinkage, and hence the update direction depends on $\alpha_k$. In contrast, the update direction of `SPS` does not depend on $\alpha_k$, and the regularization effect is intertwined with the adaptive step size. Another way to see that the model (10) on which `ProxSPS` is based on is a more accurate model as compared to the `SPS` model (9), is that the resulting vector field of `ProxSPS` takes a more direct route to the minimum, as illustrated in Fig. 1b.

Update (14) needs to compute the term $\langle\nabla f_{i_k}(x^k), x^k\rangle$ while (13) needs to evaluate $\|x^k\|^2$. Other than that, the computational costs are roughly identical. For (14), a lower bound $\underline{\ell}_i$ is required. For non-negative loss

functions, in practice both $\underline{\ell}_i$ and $C(s_i)$ are often set to zero, in which case (10) will be a more accurate model as compared to (9). [4]

### 4.3 Convergence analysis

For the convergence analysis of Algorithm 2, we can work with the following assumption on $\varphi$.

**Assumption 3.** $\varphi : \mathbb{R}^n \to \mathbb{R} \cup \{\infty\}$ *is a proper, closed, $\lambda$-strongly convex function with $\lambda \geq 0$.*

Throughout this section we consider model (10), i.e. for $g \in \partial f(x; s)$, let

$$\psi_x(y; s) = f_x(y; s) + \varphi(y), \quad f_x(y; s) = \max\{f(x; s) + \langle g, y - x \rangle, C(s)\}.$$

Let us first state a lemma on important properties of the truncated model:

**Lemma 2.** *Consider $f_x(y; s) = \max\{f(x; s) + \langle g, y - x \rangle, C(s)\}$, where $g \in \partial f(x; s)$ is arbitrary and $C(s) \leq \inf_{z \in \mathbb{R}^n} f(z; s)$. Then, it holds:*

*(i) The mapping $y \mapsto f_x(y; s)$ is convex.*

*(ii) For all $x \in \mathbb{R}^n$, it holds $f_x(x; s) = f(x; s)$. If $f(\cdot; s)$ is $\rho_s$-weakly convex for all $s \in \mathcal{S}$, then*

$$f_x(y; s) \leq f(y; s) + \tfrac{\rho_s}{2}\|y - x\|^2 \quad \text{for all } x, y \in \mathbb{R}^n.$$

*Proof.*     (i) The maximum over a constant and linear term is convex.

(ii) Recall that $C(s) \leq f(y; s)$ for all $y \in \mathbb{R}^n$. Therefore, $f_x(x; s) = \max\{C(s), f(x; s)\} = f(x; s)$. From weak convexity of $f(\cdot; s)$ it follows $f(x; s) + \langle g, y - x \rangle \leq f(y; s) + \tfrac{\rho_s}{2}\|y - x\|^2$ and therefore

$$f_x(y; s) \leq \max\{C(s), f(y; s) + \tfrac{\rho_s}{2}\|y - x\|^2\} = f(y; s) + \tfrac{\rho_s}{2}\|y - x\|^2 \quad \text{for all } y \in \mathbb{R}^n.$$

$\square$

#### 4.3.1 Globally bounded subgradients

In this section, we show that the results for stochastic model-based proximal point methods in Davis & Drusvyatskiy (2019) can be immediately applied to our specific model – even though this model has not been explicitly analyzed in their article. This, however, requires assuming that the subgradients are bounded.

**Proposition 3.** *Let Assumption 3 hold and assume that there is an open, convex set $U$ containing $\mathrm{dom}\,\varphi$. Let $f(\cdot; s)$ be $\rho_s$-weakly convex for all $s \in \mathcal{S}$ and let $\rho = \mathbb{E}[\rho_S]$. Assume that there exists $G_s \in \mathbb{R}_+$ for all $s \in \mathcal{S}$, such that $\mathsf{G} := \sqrt{\mathbb{E}[G_S^2]} < \infty$ and*

$$\|g(x; s)\| \leq G_s \quad \forall g(x; s) \in \partial f(x; s), \ \forall x \in U. \tag{15}$$

*Then, $\psi_x(y; s)$ satisfies (Davis & Drusvyatskiy, 2019, Assum. B), in particular it holds*

$$f_x(x; s) - f_x(y; s) \leq G_s \|x - y\| \quad \text{for all } s \in \mathcal{S} \text{ and all } x, y \in U. \tag{16}$$

**Remark 1.** *We state all four properties (B1)–(B4) of (Davis & Drusvyatskiy, 2019, Assum. B) explicitly in the Appendix, see Proposition 14 which also contains the proof. The first three properties follow immediately in our setting. Only the last property (B4), stated in (16), requires the additional assumption (15).*

**Corollary 4** (Weakly convex case)**.** *Let the assumptions of Proposition 3 hold with $\rho_s > 0$ for all $s \in \mathcal{S}$. Let $\rho = \mathbb{E}[\rho_S] < \infty$ and let $\Delta \geq \mathrm{env}_\psi^{1/(2\rho)}(x^0) - \min \psi$. Let $\{x^k\}_{k=0,\dots,K}$ be generated by Algorithm 2 for constant step sizes $\alpha_k = \left(2\rho + \sqrt{\frac{4\rho \mathsf{G}^2 K}{\Delta}}\right)^{-1}$. Then, it holds*

$$\mathbb{E}\|\nabla \mathrm{env}_\psi^{1/(2\rho)}(x_\sim^K)\|^2 \leq \frac{8\rho\Delta}{K} + 16\mathsf{G}\sqrt{\frac{\rho\Delta}{K}},$$

*where $x_\sim^K$ is uniformly drawn from $\{x^0, \dots, x^{K-1}\}$.*

---

[4]For single element sampling, $\inf \ell_i$ can sometimes be precomputed (e.g. regularized logistic regression, see (Loizou et al., 2021, Appendix D)). But even in this restricted setting it is not clear how to estimate $\inf \ell_i$ when using mini-batching.

*Proof.* The claim follows from Proposition 3 and (Davis & Drusvyatskiy, 2019, Thm. 4.3), (4.16) setting $\eta = 0$, $\bar{\rho} = 2\rho$, $T = K - 1$ and $\beta_t = \alpha_k^{-1}$. □

**Corollary 5** ((Strongly) convex case)**.** *Let the assumptions of Proposition 3 hold with $\rho_s = 0$ for all $s \in \mathcal{S}$. Let $\lambda > 0$ and $x^\star = \arg\min_x \psi(x)$. Let $\{x^k\}_{k=0,\dots,K}$ be generated by Algorithm 2 for step sizes $\alpha_k = \frac{2}{\lambda(k+1)}$. Then, it holds*

$$\mathbb{E}\Big[\psi\Big(\tfrac{2}{(K+1)(K+2)-2}\sum_{k=1}^{K}(k+1)x^k\Big) - \psi(x^\star)\Big] \leq \frac{\lambda}{(K+1)^2}\|x^0 - x^\star\|^2 + \frac{8\mathsf{G}^2}{\lambda(K+1)}.$$

*Proof.* As $\rho_s = 0$ and hence $\rho = 0$, we have that (Davis & Drusvyatskiy, 2019, Assum. B) is satisfied with $\tau = 0$ (in the notation of (Davis & Drusvyatskiy, 2019), see Proposition 14). Moreover, by Lemma 2, (i) and $\lambda$–strong convexity of $\varphi$, we have $\lambda$–strong convexity of $\psi_x(\cdot; s)$. The claim follows from Proposition 3 and (Davis & Drusvyatskiy, 2019, Thm. 4.5) setting $\mu = \lambda$, $T = K - 1$ and $\beta_t = \alpha_k^{-1}$. □

### 4.3.2 Lipschitz smoothness

Assumption (15), i.e. having globally bounded subgradients, is strong: it implies Lipschitz continuity of $f$ (cf. (Davis & Drusvyatskiy, 2019, Lem. 4.1)) and simple functions such as the squared loss do not satisfy this. Therefore, we provide additional guarantees for the smooth case, without the assumption of globally bounded gradients.

The following result, similar to (Davis & Drusvyatskiy, 2019, Lem. 4.2), is the basic inequality for the subsequent convergence analysis.

**Lemma 6.** *Let Assumption 3 hold. Let $x^{k+1}$ be given by (11) and $\psi_{x^k}$ be given in (10). For every $x \in \mathbb{R}^n$ it holds*

$$(1 + \alpha_k\lambda)\|x^{k+1} - x\|^2 \leq \|x^k - x\|^2 - \|x^{k+1} - x^k\|^2 + 2\alpha_k\big(\psi_{x^k}(x; S_k) - \psi_{x^k}(x^{k+1}; S_k)\big). \tag{17}$$

*Moreover, it holds*

$$\psi_{x^k}(x^{k+1}; S_k) \geq f(x^k; S_k) + \langle g_k, x^{k+1} - x^k\rangle + \varphi(x^{k+1}). \tag{18}$$

*Proof.* The objective of (11) is given by $\Psi_k(y) := \psi_{x^k}(y; S_k) + \frac{1}{2\alpha_k}\|y - x^k\|^2$. Using Lemma 2, (i) and $\lambda$-strong convexity of $\varphi$, $\Psi_k(y)$ is $(\lambda + \frac{1}{\alpha_k})$–strongly convex. As $x^{k+1}$ is the minimizer of $\Psi_k(y)$, for all $x \in \mathbb{R}^n$ we have

$$\Psi_k(x) \geq \Psi_k(x^{k+1}) + \frac{1 + \alpha_k\lambda}{2\alpha_k}\|x^{k+1} - x\|^2 \iff$$
$$(1 + \alpha_k\lambda)\|x^{k+1} - x\|^2 \leq \|x^k - x\|^2 - \|x^{k+1} - x^k\|^2 + 2\alpha_k\big(\psi_{x^k}(x; S_k) - \psi_{x^k}(x^{k+1}; S_k)\big).$$

Moreover, by definition of $f_x(y; s)$ in (10) we have

$$\psi_{x^k}(x^{k+1}; S_k) = f_{x^k}(x^{k+1}; S_k) + \varphi(x^{k+1}) \geq f(x^k; S_k) + \langle g_k, x^{k+1} - x^k\rangle + \varphi(x^{k+1}).$$

□

We will work in the setting of differentiable loss functions with bounded gradient noise.

**Assumption 4.** *The mapping $f(\cdot; s)$ is differentiable for all $s \in \mathcal{S}$ and there exists $\beta \geq 0$ such that*

$$\mathbb{E}\|\nabla f(x; S) - \nabla f(x)\|^2 \leq \beta \quad \text{for all } x \in \mathbb{R}^n. \tag{19}$$

The assumption of bounded gradient noise (19) (in the differentiable setting) is indeed a weaker assumption than (15) since $\mathbb{E}[\nabla f(x; S)] = \nabla f(x)$ and

$$\mathbb{E}\|\nabla f(x; S) - \nabla f(x)\|^2 \leq \beta \iff \mathbb{E}\|\nabla f(x; S)\|^2 \leq \|\nabla f(x)\|^2 + \beta.$$

**Remark 2.** *Assumption 4 (and the subsequent theorems) could be adapted to the case where $f(\cdot; s)$ is weakly convex but non-differentiable: for fixed $x \in \mathbb{R}^n$, due to (Bertsekas, 1973, Prop. 2.2) and (Davis & Drusvyatskiy, 2019, Lem. 2.1) it holds*

$$\mathbb{E}[\partial f(x; S)] = \mathbb{E}\Big[\partial\big(f(x; S) + \frac{\rho_S}{2}\|x\|^2\big) - \rho_S x\Big] = \partial f(x) + \rho x - \mathbb{E}[\rho_S x] = \partial f(x),$$

*where we used $\rho = \mathbb{E}[\rho_S]$. Hence, for $g_s \in \partial f(x; s)$ we have $\mathbb{E}[g_S] \in \partial f(x)$ and (19) is replaced by*

$$\mathbb{E}\|g_S - \mathbb{E}[g_S]\|^2 \le \beta \quad \text{for all } x \in \mathbb{R}^n.$$

*However, as we will still require that $f$ is Lipschitz-smooth, we present our results for the differentiable setting.*

The proof of the subsequent theorems can be found in Appendix A.2 and Appendix A.3.

**Theorem 7.** *Let Assumption 3 and Assumption 4 hold. Let $f(\cdot; s)$ be convex for all $s \in \mathcal{S}$ and let $f$ be $L$–smooth (4). Let $x^\star = \arg\min_{x \in \mathbb{R}^n} \psi(x)$ and let $\theta > 1$. Let $\{x^k\}_{k=0,\dots,K}$ be generated by Algorithm 2 for step sizes $\alpha_k > 0$ such that*

$$\alpha_k \le \frac{1 - 1/\theta}{L}. \tag{20}$$

*Then, it holds*

$$(1 + \alpha_k \lambda)\mathbb{E}\|x^{k+1} - x^\star\|^2 \le \mathbb{E}\|x^k - x^\star\|^2 + 2\alpha_k \mathbb{E}[\psi(x^\star) - \psi(x^{k+1})] + \theta\beta\alpha_k^2. \tag{21}$$

*Moreover, we have:*

    *a) If $\lambda > 0$ and $\alpha_k = \frac{1}{\lambda(k + k_0)}$ with $k_0 \ge 1$ large enough such that (20) is fulfilled, then*

$$\mathbb{E}\Big[\psi\Big(\frac{1}{K}\sum_{k=0}^{K-1} x^{k+1}\Big) - \psi(x^\star)\Big] \le \frac{\lambda k_0}{2K}\|x^0 - x^\star\|^2 + \frac{\theta\beta(1 + \ln K)}{2\lambda K}. \tag{22}$$

    *b) If $\lambda = 0$ and $\alpha_k = \frac{\alpha}{\sqrt{k+1}}$ with $\alpha \le \frac{1 - 1/\theta}{L}$, then*

$$\mathbb{E}\Big[\psi\Big(\frac{1}{\sum_{k=0}^{K-1}\alpha_k}\sum_{k=0}^{K-1}\alpha_k x^{k+1}\Big) - \psi(x^\star)\Big] \le \frac{\|x^0 - x^\star\|^2}{4\alpha(\sqrt{K+1} - 1)} + \frac{\theta\beta\alpha(1 + \ln K)}{4(\sqrt{K+1} - 1)}. \tag{23}$$

    *c) If $f$ is $\mu$–strongly convex with $\mu \ge 0$,[5] and $\alpha_k = \alpha$ fulfilling (20), then*

$$\mathbb{E}\|x^K - x^\star\|^2 \le (1 + \alpha(\mu + 2\lambda))^{-K}\|x^0 - x^\star\|^2 + \frac{\theta\beta\alpha}{\mu + 2\lambda}. \tag{24}$$

**Remark 3.** *If $\lambda > 0$, for the decaying step sizes in item a) we get a rate of $\tilde{\mathcal{O}}(\frac{1}{K})$ if $\lambda > 0$. In the strongly convex case in item c), for constant step sizes, we get a linear convergence upto a neighborhood of the solution. Note that the constant on the right-hand side of (24) can be forced to be small using a small $\alpha$. Further, the rate (24) has a $2\lambda$ term, instead of $\lambda$. This slight improvement in the rate occurs because we do not linearize $\varphi$ in the* `ProxSPS` *model.*

**Theorem 8.** *Let Assumption 3 and Assumption 4 hold. Let $f(\cdot; s)$ be $\rho_s$–weakly convex for all $s \in \mathcal{S}$ and let $\rho := \mathbb{E}[\rho_S] < \infty$. Let $f$ be $L$–smooth[6] and assume that $\inf \psi > -\infty$. Let $\{x^k\}_{k \ge 0}$ be generated by Algorithm 2. For $\theta > 1$, under the condition*

$$\eta \in \begin{cases} (0, \frac{1}{\rho - \lambda}) & \text{if } \rho > \lambda, \\ (0, \infty) & \text{else} \end{cases}, \qquad \alpha_k \le \frac{1 - \theta^{-1}}{L + \eta^{-1}}, \tag{25}$$

---

[5]Note that as $f(\cdot; s)$ is convex, so is $f$, and that we allow $\mu = 0$ here.
[6]As $f$ is $\rho$–weakly convex, this implies $\rho \le L$.

*it holds*

$$\sum_{k=0}^{K-1} \alpha_k \mathbb{E}\|\nabla \mathrm{env}_\psi^\eta(x^k)\|^2 \leq \frac{2(\mathrm{env}_\psi^\eta(x^0) - \inf \psi)}{1 - \eta(\rho - \lambda)} + \frac{\beta\theta}{\eta(1 - \eta(\rho - \lambda))} \sum_{k=0}^{K-1} \alpha_k^2. \tag{26}$$

*Moreover, for the choice $\alpha_k = \frac{\alpha}{\sqrt{k+1}}$ and with $\alpha \leq \frac{1-\theta^{-1}}{L+\eta^{-1}}$, we have*

$$\min_{k=0,\dots,K-1} \mathbb{E}\|\nabla \mathrm{env}_\psi^\eta(x^k)\|^2 \leq \frac{\mathrm{env}_\psi^\eta(x^0) - \inf \psi}{\alpha(1 - \eta(\rho - \lambda))(\sqrt{K+1} - 1)} + \frac{\beta\theta}{2\eta(1 - \eta(\rho - \lambda))} \frac{\alpha(1 + \ln K)}{(\sqrt{K+1} - 1)}.$$

*If instead we choose $\alpha_k = \frac{\alpha}{\sqrt{K}}$ and with $\alpha \leq \sqrt{K}\frac{1-\theta^{-1}}{L+\eta^{-1}}$, we have*

$$\mathbb{E}\|\nabla \mathrm{env}_\psi^\eta(x_\sim^K)\|^2 \leq \frac{2(\mathrm{env}_\psi^\eta(x^0) - \inf \psi)}{\alpha(1 - \eta(\rho - \lambda))\sqrt{K}} + \frac{\beta\theta}{\eta(1 - \eta(\rho - \lambda))} \frac{\alpha}{\sqrt{K}},$$

*where $x_\sim^K$ is uniformly drawn from $\{x^0, \dots, x^{K-1}\}$.*

### 4.3.3 Comparison to existing theory

Recalling that Algorithm 1 is equivalent to $\mathtt{SPS_{max}}$ with $c = 1$ and $\gamma_b = \alpha_k$, we can apply Theorem 7 and Theorem 8 for the unregularized case where $\varphi = 0$ and hence obtain new theory for ($\mathtt{SPS_{max}}$). We start by summarizing the main theoretical results for $\mathtt{SPS_{max}}$ given in (Loizou et al., 2021; Orvieto et al., 2022): in the (ER) setting, recall the interpolation constant $\sigma^2 = \mathbb{E}[f(x^\star; S) - C(S)] = \frac{1}{N}\sum_{i=1}^N f_i(x^\star) - C(s_i)$. If $f_i$ is $L_i$-smooth and convex, (Orvieto et al., 2022, Thm. 3.1) proves convergence to a neighborhood of the solution, i.e. the iterates $\{x^k\}$ of $\mathtt{SPS_{max}}$ satisfy

$$\mathbb{E}[f(\bar{x}^K) - f(x^\star)] \leq \frac{\|x^0 - x^\star\|^2}{\alpha K} + \frac{2\gamma_b\sigma^2}{\alpha}, \tag{27}$$

where $\bar{x}^K := \frac{1}{K}\sum_{k=0}^{K-1} x^k$, $\alpha := \min\{\frac{1}{2cL_{\max}}, \gamma_b\}$, and $L_{\max} := \max_{i \in [N]} L_i$.[7] For the nonconvex case, if $f_i$ is $L_i$-smooth and under suitable assumptions on the gradient noise, (Loizou et al., 2021, Thm. 3.8) states that, for constants $c_1$ and $c_2$, we have

$$\min_{k=1,\dots,K} \mathbb{E}\|\nabla f(x^k)\|^2 \leq \frac{1}{c_1 K} + c_2. \tag{28}$$

The main advantage of these results is that $\gamma_b$ can be held constant; furthermore in the convex setting (27), the choice of $\gamma_b$ requires no knowledge of the smoothness constants $L_i$. For both results however, we can not directly conclude that the right-hand side goes to zero as $K \to \infty$ as there is an additional constant. Choosing $\gamma_b$ sufficiently small does not immediately solve this as $c_1$, $\alpha$ and $c_2$ all go to zero as $\gamma_b$ goes to zero.

Our results complement this by showing exact convergence for the (weakly) convex case, i.e. without constants on the right-hand side. This comes at the cost of an upper bound on the step sizes $\alpha_k$ which depends on the smoothness constant $L$. For exact convergence, it is important to use decreasing step sizes $\alpha_k$: Theorem 8 shows that the gradient of the Moreau envelope converges to zero at the rate $\mathcal{O}(1/\sqrt{K})$ for the choice of $\alpha_k = \frac{\alpha}{\sqrt{K}}$.[8] Another minor difference to (Loizou et al., 2021) is that we do not need to assume Lipschitz-smoothness for all $f(\cdot; s)$ and work instead with the (more general) assumption of weak convexity. However, we still need to assume Lipschitz smoothness of $f$.

Another variant of $\mathtt{SPS_{max}}$, named DecSPS, has been proposed in (Orvieto et al., 2022): for unregularized problems (1) it is given by

$$x^{k+1} = x^k - \hat{\gamma}_k g_k, \quad \hat{\gamma}_k = \frac{1}{c_k} \min\left\{\frac{f(x^k; S_k) - C_k}{\|g_k\|^2}, c_{k-1}\hat{\gamma}_{k-1}\right\} \tag{DecSPS}$$

---

[7]The theorem also handles the mini-batch case but, for simplicity, we state the result for sampling a single $i_k$ in each iteration.

[8]Notice that $\alpha_k$ then depends on the total number of iterations $K$ and hence one would need to fix $K$ before starting the method.

where $\{c_k\}_{k\geq 0}$ is an increasing sequence. In the (ER) setting, if all $f_i$ are Lipschitz-smooth and strongly convex, `DecSPS` converges with a rate of $\mathcal{O}(\frac{1}{\sqrt{K}})$, without knowledge of the smoothness or convexity constants (cf. (Orvieto et al., 2022, Thm. 5.5)). However, under these assumptions, the objective $f$ is strongly convex and the optimal rate is $\mathcal{O}(\frac{1}{K})$, which we achieve up to a logarithmic factor in Theorem 7, (22). Moreover, for `DecSPS` no guarantees are given for nonconvex problems.

For regularized problems, the constant in (27) is problematic if $\sigma^2$ (computed for the regularized loss) is moderately large. We refer to Appendix D.5 where we show that this can easily happen. For `ProxSPS`, our theoretical results Theorem 7 and Theorem 8 are not affected by this as they do not depend on the size of $\sigma^2$. To the best of our knowledge, this is the first work to show theory for the stochastic Polyak step size in a setting that explicitly considers regularization. Moreover, our results also cover the case of non-smooth or non-real-valued regularization $\varphi$ where the theory in (Loizou et al., 2021) can not be applied.

## 5 Numerical experiments

Throughout we denote Algorithm 1 with `SPS` and Algorithm 3 with `ProxSPS`. For all experiments we use `PyTorch` (Paszke et al., 2019)[9].

### 5.1 General parameter setting

For `SPS` and `ProxSPS` we always use $C(s) = 0$ for all $s \in \mathcal{S}$. For $\alpha_k$, we use the following schedules:

- `constant`: set $\alpha_k = \alpha_0$ for all $k$ and some $\alpha_0 > 0$.

- `sqrt`: set $\alpha_k = \frac{\alpha_0}{\sqrt{j}}$ for all iterations $k$ during epoch $j$.

As we consider problems with $\ell_2$-regularization, for `SPS` we handle the regularization term by incorporating it into all individual loss functions, as depicted in (13). With $\varphi = \frac{\lambda}{2}\|\cdot\|^2$ for $\lambda \geq 0$, we denote by $\zeta_k$ the *adaptive step size* term of the following algorithms:

- for `SPS` we have $\zeta_k := \frac{f(x^k;S_k)+\frac{\lambda}{2}\|x^k\|^2}{\|g_k+\lambda x^k\|^2}$ (cf. (13) with $\underline{\ell}_{i_k} = 0$ ),

- for `ProxSPS` we have $\zeta_k := \left(\frac{(1+\alpha_k\lambda)f(x^k;S_k)-\alpha_k\lambda\langle g_k, x^k\rangle}{\|g_k\|^2}\right)_+$ and thus $\tau_k^+ = \min\{\alpha_k, \zeta_k\}$ (cf. Lemma 1 with $C(S_k) = 0$).

### 5.2 Regularized matrix factorization

**Problem description:** For $A \in \mathbb{R}^{q\times p}$, consider the problem

$$\min_{W_1\in\mathbb{R}^{r\times p}, W_2\mathbb{R}^{q\times r}} \mathbb{E}_{y\sim N(0,I)}\|W_2W_1y - Ay\|^2 = \min_{W_1\in\mathbb{R}^{r\times p}, W_2\mathbb{R}^{q\times r}}\|W_2W_1 - A\|_F^2.$$

For the above problem, $\text{SPS}_{\max}$ has shown superior performance than other methods in the numerical experiments of (Loizou et al., 2021). The problem can can be turned into a (nonconvex) empirical risk minimization problem by drawing $N$ samples $\{y^{(1)}, \ldots, y^{(N)}\}$. Denote $b^{(i)} := Ay^{(i)}$. Adding squared norm regularization with $\lambda \geq 0$ (cf. (Srebro et al., 2004)), we obtain the problem

$$\min_{W_1\in\mathbb{R}^{r\times p}, W_2\mathbb{R}^{q\times r}} \frac{1}{N}\sum_{i=1}^{N}\|W_2W_1y^{(i)} - b^{(i)}\|^2 + \frac{\lambda}{2}\left(\|W_1\|_F^2 + \|W_2\|_F^2\right). \tag{29}$$

This fits the format of (2), where $x = (W_1, W_2)$, using a finite sample space $\mathcal{S} = \{s_1, \ldots, s_N\}$, $f(x;s_i) = \|W_2W_1y^{(i)} - Ay^{(i)}\|^2$, and $\varphi = \frac{\lambda}{2}\|\cdot\|_F^2$. Clearly, zero is a lower bound of $f(\cdot;s_i)$ for all $i \in [N]$. We investigate `ProxSPS` for problems of form (29) on synthetic data. For details on the experimental procedure, we refer

---

[9]The code for our experiments and an implementation of `ProxSPS` can be found at https://github.com/fabian-sp/ProxSPS.

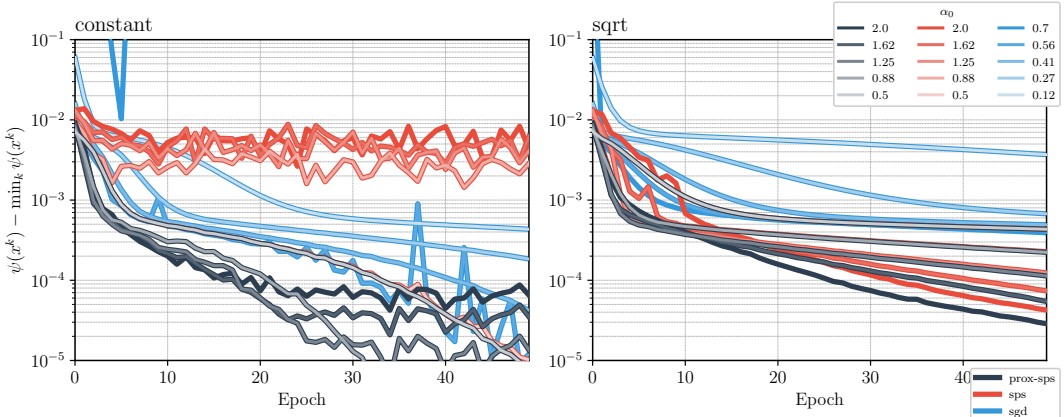

Figure 2: Objective function for the Matrix Factorization problem (29), with `constant` (left) and `sqrt` (right) step size schedule and several choices of initial values. Here $\min_k \psi(x^k)$ is the best objective function value found over all methods and all iterations.

to Appendix D.1.

**Discussion:** We discuss the results for the setting `matrix-fac1` in Table 1 in the Appendix. We first fix $\lambda = 0.001$ and consider the three methods `SPS`, `ProxSPS` and `SGD`. Fig. 2 shows the objective function over 50 epochs, for both step size schedules `sqrt` and `constant`, and several initial values $\alpha_0$. For the `constant` schedule, we observe that `ProxSPS` converges quickly for all initial values while `SPS` is unstable. Note that for `SGD` we need to pick much smaller values for $\alpha_0$ in order to avoid divergence (`SGD` diverges for large $\alpha_0$). `SPS` for large $\alpha_0$ is unstable, while for small $\alpha_0$ we can expect similar performance to `SGD` (as $\gamma_k$ is capped by $\alpha_k = \alpha_0$). However, in the regime of small $\alpha_0$, convergence will be very slow. Hence, one of the main advantages of `SPS`, namely that its step size can be chosen constant and moderately large (compared to `SGD`), is not observed here. `ProxSPS` fixes this by admitting a larger range of initial step sizes, all of which result in fast convergence, and therefore is more robust than `SGD` and `SPS` with respect to the tuning of $\alpha_0$.

For the `sqrt` schedule, we observe in Fig. 2 that `SPS` can be stabilized by reducing the values of $\alpha_k$ over the course of the iterations. However, for large $\alpha_0$ we still see instability in the early iterations, whereas `ProxSPS` does not show this behaviour. We again observe that `ProxSPS` is less sensitive with respect to the choice of $\alpha_0$ as compared to `SGD`. The empirical results also confirm our theoretical statement, showing exact convergence if $\alpha_k$ is decaying in the order of $1/\sqrt{k}$. From Fig. 3, we can make similar observations for the validation error, defined as $\frac{1}{N_{\text{val}}} \sum_{i=1}^{N_{\text{val}}} \|W_2 W_1 y^{(i)} - b_{\text{val}}^{(i)}\|^2$, where $b_{\text{val}}^{(i)}$ are the $N_{\text{val}} = N$ measurements from the validation set (cf. Appendix D.1 for details).

We now consider different values for $\lambda$ and only consider the `sqrt` schedule, as we have seen that for constant step sizes, `SPS` would not work for large step sizes and be almost identical to `SGD` for small step sizes. Fig. 4 shows the objective function and validation error. Again, we can observe that `SPS` is unstable for large initial values $\alpha_0$ for all $\lambda \geq 10^{-4}$. On the other hand, `ProxSPS` has a good performance for a wide range of $\alpha_0 \in [1, 10]$ if $\lambda$ is not too large. Indeed, `ProxSPS` convergence only starts to deteriorate when both $\alpha_0$ and $\lambda$ are very large. For $\alpha_0 = 1$, the two methods give almost identical results. Finally, in Fig. 5a we plot the validation error as a function of $\lambda$ (taking the median over the last ten epochs). The plot shows that the best validation error is obtained for $\lambda = 10^{-4}$ and for large $\alpha_0$. With `SPS` the validation error is higher, in particular for large $\alpha_0$ and $\lambda$. Fig. 5b shows that `ProxSPS` leads to smaller norm of the iterates, hence a more effective regularization. Finally, we plot the actual step sizes for both methods in Fig. 6. We observe that the adaptive step size $\zeta_k$ (Definition at end of Section 5.1) is typically larger and has more variance for `SPS` than `ProxSPS`, in particular for large $\lambda$. This increased variance might explain why `SPS` is unstable when $\alpha_0$ is large: the actual step size is the minimum between $\alpha_k$ and $\zeta_k$ and hence both terms being large could lead to instability. On the other hand, if $\alpha_0 = 1$, the plot confirms that `SPS` and `ProxSPS` are almost identical methods as $\zeta_k > \alpha_k$ for most iterations.

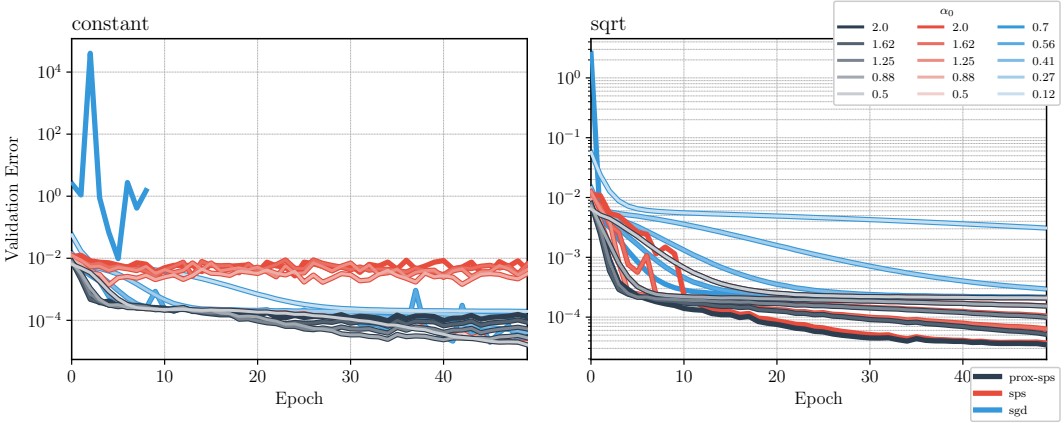

Figure 3: Validation error for the Matrix Factorization problem (29), with `constant` (left) and `sqrt` (right) step size schedule and several choices of initial values.

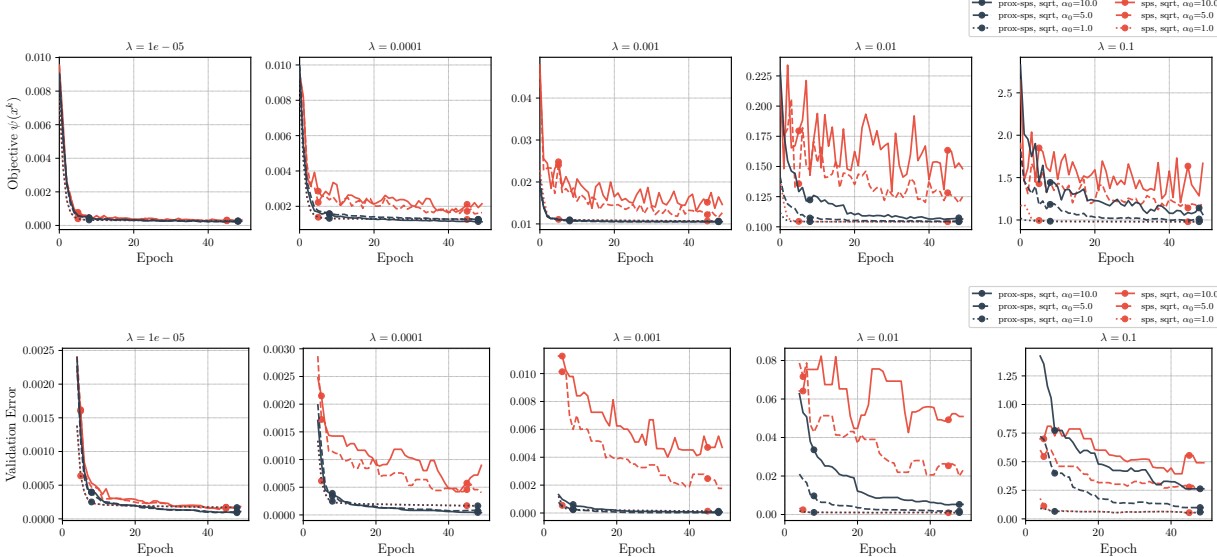

Figure 4: Objective function value and validation error over the course of optimization. For the validation error, we plot a rolling median over five epochs in order to avoid clutter.

We provide additional numerical results which confirm the above findings in the Appendix: this includes the results for the setting `matrix-fac2` of Table 1 in Appendix D.2 as well as a matrix completion task on a real-world dataset of air quality sensor networks (Rivera-Muñoz et al., 2022) in Appendix D.3.

## 5.3 Deep networks for image classification

We train a `ResNet56` and `ResNet110` model (He et al., 2016) on the `CIFAR10` dataset. We use the data loading and preprocessing procedure and network implementation from https://github.com/akamaster/pytorch_resnet_cifar10. We do not use batch normalization. The loss function is the cross-entropy loss of the true image class with respect to the predicted class probabilities, being the output of the `ResNet56` network. We add $\frac{\lambda}{2}\|x\|^2$ as regularization term, where $x$ consists of all learnable parameters of the model. The `CIFAR10` dataset consists of 60,000 images, each of size $32 \times 32$, from ten different classes. We use the `PyTorch` split into 50,000 training and 10,000 test examples and use a batch size of 128. For `AdamW`, we set the weight decay parameter to $\lambda$ and set all other hyperparameters to its default. We use the

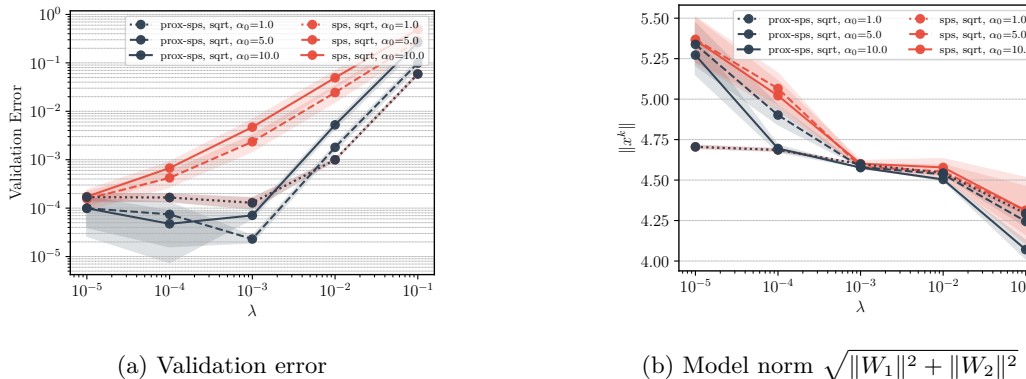

(a) Validation error

(b) Model norm $\sqrt{\|W_1\|^2 + \|W_2\|^2}$

Figure 5: Validation error and model norm as a function of the regularization parameter $\lambda$. Shaded area is one standard deviation (computed over ten independent runs). For all values, we take the median over epochs $[40, 50]$.

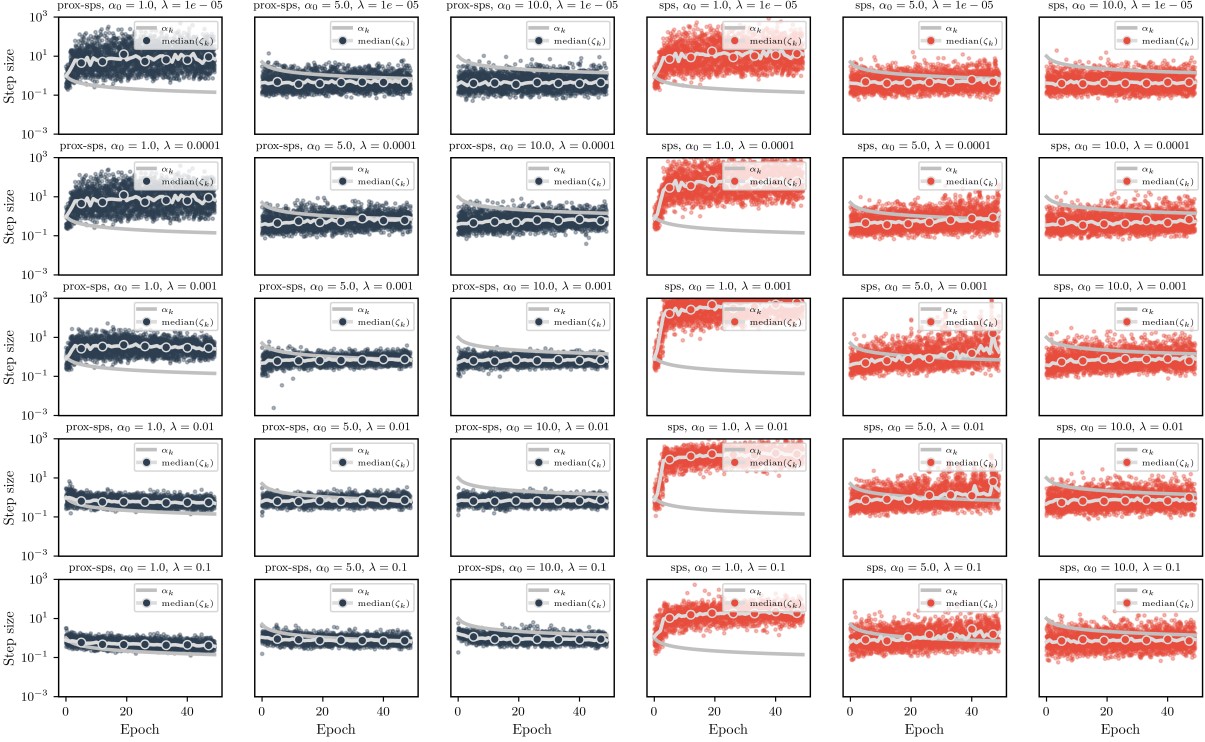

Figure 6: Adaptive step size selection for `SPS` and `ProxSPS`. We plot $\zeta_k$ (see definition in Section 5.1) as dots for each iteration as well as their median over each epoch. For this plot, we use the results of only one of the ten runs.

`AdamW`-implementation from https://github.com/zhenxun-zhuang/AdamW-Scale-free as it does not – in contrast to the `Pytorch` implementation – multiply the weight decay parameter with the learning rate, which leads to better comparability to `SPS` and `ProxSPS` for identical values of $\lambda$. For `SPS` and `ProxSPS` we use the `sqrt`-schedule and $\alpha_0 = 1$. We run each method repeatedly using (the same) three different seeds for the dataset shuffling.

**Discussion:** For `Resnet56`, from the bottom plot in Fig. 7, we observe that both `SPS` and `ProxSPS` work well

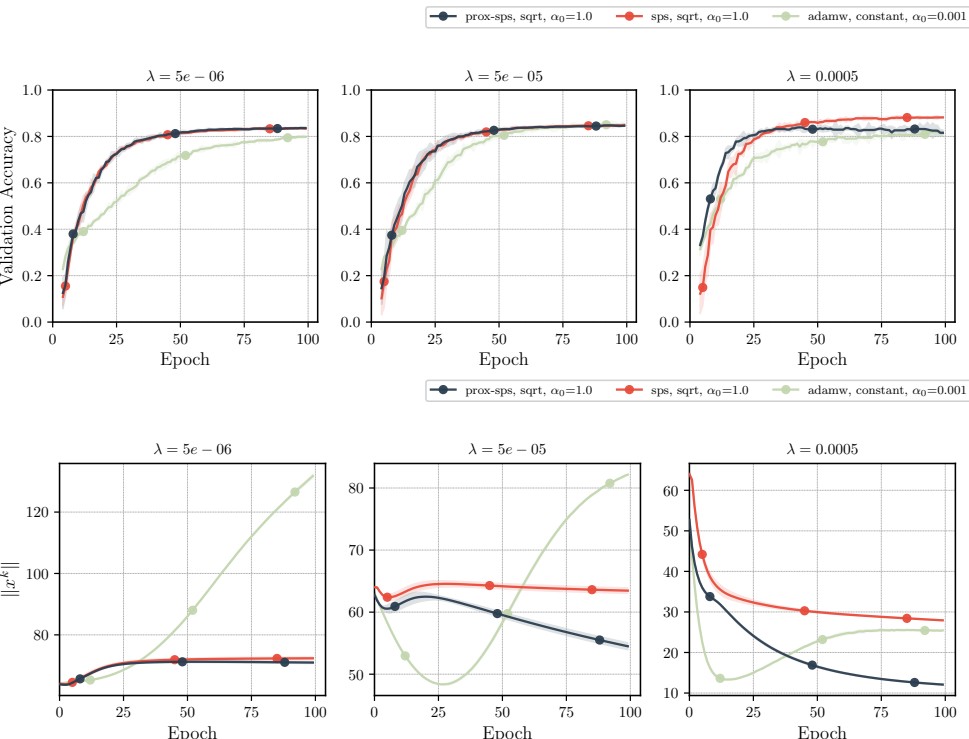

Figure 7: `ResNet56`: (Top): Validation accuracy and model norm for three values of the regularization parameter $\lambda$. Validation accuracy is defined as the ratio of correctly labeled images on the validation set (i.e. *Top-1 accuracy*), plotted as five-epoch running median. (Bottom): With $\|x^k\|$ we denote the norm of all learnable parameters at the $k$-th iteration. Shaded area is two standard deviations over three independent runs.

with `ProxSPS` leading to smaller weights. For $\lambda = 5e - 4$, the progress of `ProxSPS` stagnates after roughly 25 epochs. This can be explained by looking at the adaptive step size term $\zeta_k$ in Fig. 9a: as it decays over time we have $\tau_k^+ = \zeta_k \ll \alpha_k$. Since every iteration of `ProxSPS` shrinks the weights by a factor $\frac{1}{1+\alpha_k\lambda}$, this leads to a bias towards zero. This suggests that we should choose $\alpha_k$ roughly of the order of $\zeta_k$, for example by using the values of $\zeta_k$ from the previous epoch.

For the larger model `Resnet110` however, `SPS` does not make progress for a long time because the adaptive step size is very small (see Fig. 8 and Fig. 9b). `ProxSPS` does not share this issue and performs well after a few initial epochs. For larger values of $\lambda$, the training is also considerably faster than for `AdamW`. Generally, we observe that `ProxSPS` (and `SPS` for `Resnet56`) performs well in comparison to `AdamW`. This is achieved without extensive hyperparameter tuning (in particular this suggests that setting $c = 1$ in $\text{SPS}_{\max}$ leads to good results and reduces tuning effort).

Furthermore, we trained a `ResNet110` *with* batch norm on the `Imagenet32` dataset. The plots and experimental details can be found in Appendix D.4. From Fig. 15, we conclude that `SPS` and `ProxSPS` perform equally well in this experiment. Both `SPS` and `ProxSPS` are less sensititve with respect to the regularization parameter $\lambda$ than `AdamW` and the adaptive step size leads to faster learning in the initial epochs compared to `SGD`. We remark that with batch norm, the effect of $\ell_2$-regularization is still unclear as the output of batch norm layers is invariant to scaling and regularization becomes ineffective (Zhang et al., 2019).

## 6 Conclusion

We proposed and analyzed `ProxSPS`, a proximal version of the stochastic Polyak step size. We arrived at `ProxSPS` by using the framework of stochastic model-based proximal point methods. We then used this framework to argue that the resulting model of `ProxSPS` is a better approximation as compared to the model

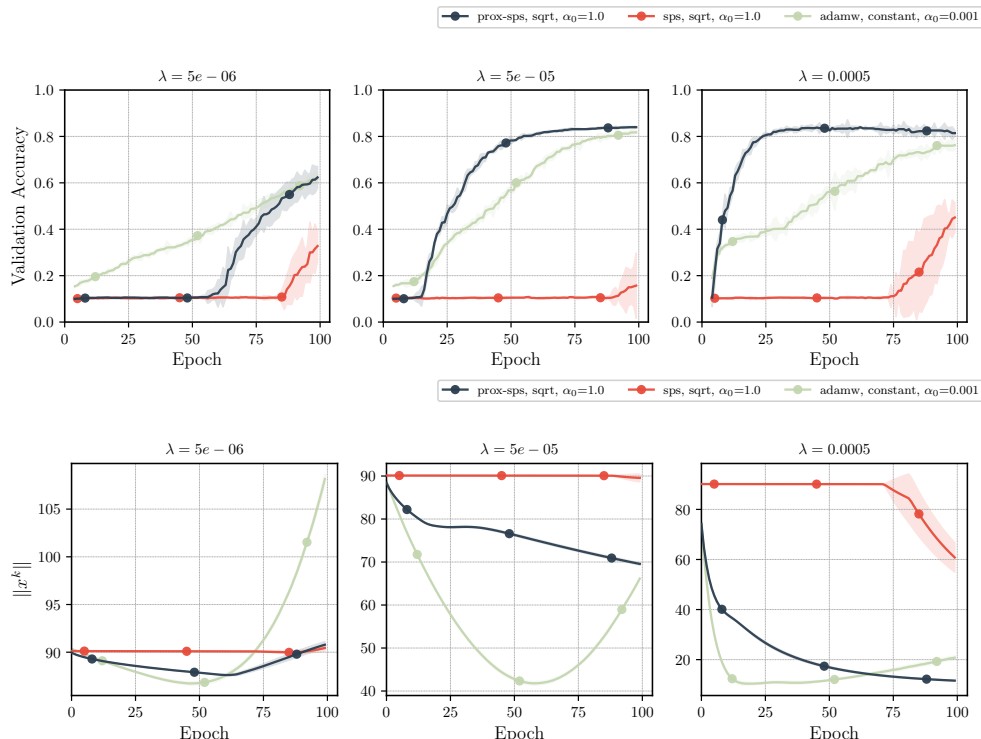

Figure 8: `ResNet110`: Validation accuracy as five-epoch running median (top) and model norm (bottom) for three values of $\lambda$. Shaded area is two standard deviations over three independent runs.

used by `SPS` when using regularization. Our theoretical results cover a wide range of optimization problems, including convex and nonconvex settings. We performed a series of experiments comparing `ProxSPS`, `SPS`, `SGD` and `AdamW` when using $\ell_2$-regularization. In particular, we find that `SPS` can be very hard to tune when using $\ell_2$-regularization, and in contrast, `ProxSPS` performs well for a wide choice of step sizes and regularization parameters. Finally, for our experiments on image classification, we find that `ProxSPS` is competitive to `AdamW`, whereas `SPS` can fail for larger models. At the same time `ProxSPS` produces smaller weights in the trained neural network. Having small weights may help reduce the memory footprint of the resulting network, and even suggests which weights can be pruned.

### Acknowledgments

We thank the Simons Foundation for hosting Fabian Schaipp at the Flatiron Institute. We also thank the TUM Graduate Center for their financial support for the visit.

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

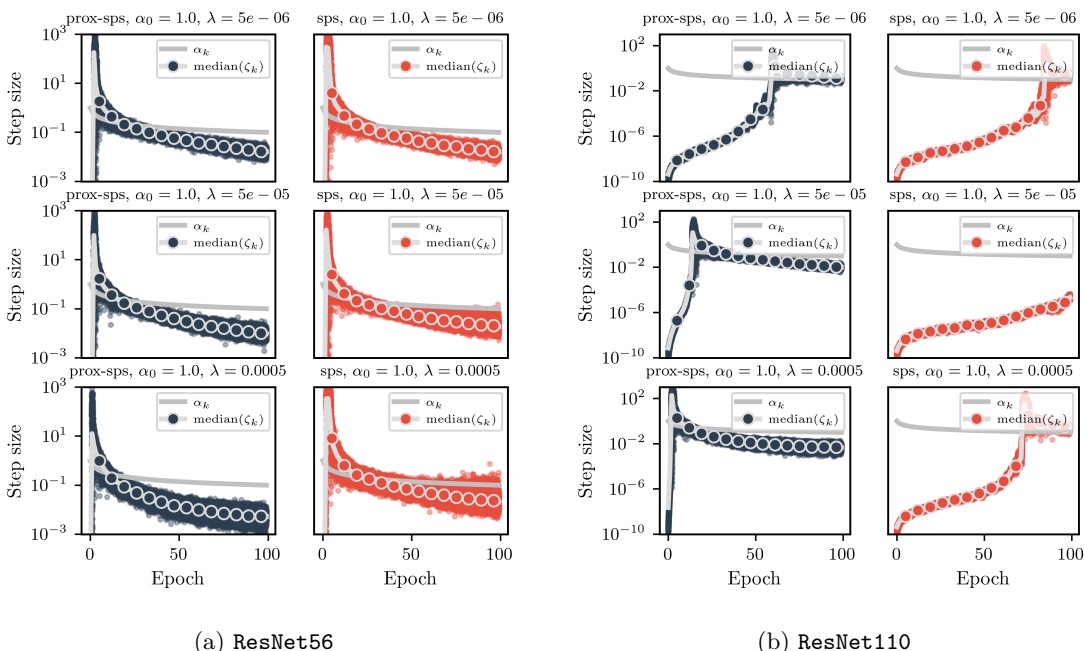

(a) `ResNet56`                    (b) `ResNet110`

Figure 9: Adaptive step sizes for `SPS` and `ProxSPS`. See definition of $\zeta_k$ in Section 5.1. For this plot, we use the results of only one of the three runs.

Léon Bottou. Large-scale machine learning with stochastic gradient descent. In *Proceedings of COMP-STAT'2010*, pp. 177–186. Physica-Verlag/Springer, Heidelberg, 2010.

Léon Bottou, Frank E. Curtis, and Jorge Nocedal. Optimization methods for large-scale machine learning. *SIAM Review*, 60(2):223–311, 2018. ISSN 0036-1445. doi: 10.1137/16M1080173.

Frank H. Clarke. *Optimization and nonsmooth analysis*. Canadian Mathematical Society Series of Monographs and Advanced Texts. John Wiley & Sons, Inc., New York, 1983. ISBN 0-471-87504-X. A Wiley-Interscience Publication.

Damek Davis and Dmitriy Drusvyatskiy. Stochastic model-based minimization of weakly convex functions. *SIAM Journal on Optimization*, 29(1):207–239, 2019. ISSN 1052-6234. doi: 10.1137/18M1178244.

Dmitriy Drusvyatskiy and Courtney Paquette. Efficiency of minimizing compositions of convex functions and smooth maps. *Mathematical Programming*, 178(1-2, Ser. A):503–558, 2019. ISSN 0025-5610. doi: 10.1007/s10107-018-1311-3.

Robert Gower, Othmane Sebbouh, and Nicolas Loizou. SGD for structured nonconvex functions: Learning rates, minibatching and interpolation. In Arindam Banerjee and Kenji Fukumizu (eds.), *Proceedings of The 24th International Conference on Artificial Intelligence and Statistics*, volume 130 of *Proceedings of Machine Learning Research*, pp. 1315–1323. PMLR, 13–15 Apr 2021. URL https://proceedings.mlr.press/v130/gower21a.html.

Elad Hazan and Sham Kakade. Revisiting the Polyak step size. May 2019.

Kaiming He, Xiangyu Zhang, Shaoqing Ren, and Jian Sun. Deep residual learning for image recognition. In *2016 IEEE Conference on Computer Vision and Pattern Recognition (CVPR)*, pp. 770–778, 2016. doi: 10.1109/CVPR.2016.90.

Diederik P. Kingma and Jimmy Ba. Adam: A method for stochastic optimization. In Yoshua Bengio and Yann LeCun (eds.), *3rd International Conference on Learning Representations, ICLR 2015, San Diego, CA, USA, May 7-9, 2015, Conference Track Proceedings*, 2015.

Nicolas Loizou, Sharan Vaswani, Issam Hadj Laradji, and Simon Lacoste-Julien. Stochastic Polyak stepsize for SGD: An adaptive learning rate for fast convergence. In Arindam Banerjee and Kenji Fukumizu (eds.), *Proceedings of The 24th International Conference on Artificial Intelligence and Statistics*, volume 130 of *Proceedings of Machine Learning Research*, pp. 1306–1314. PMLR, 13–15 Apr 2021. URL https://proceedings.mlr.press/v130/loizou21a.html.

Ilya Loshchilov and Frank Hutter. Decoupled weight decay regularization. In *7th International Conference on Learning Representations, ICLR 2019, New Orleans, LA, USA, May 6-9, 2019*. OpenReview.net, 2019. URL https://openreview.net/forum?id=Bkg6RiCqY7.

Antonio Orvieto, Simon Lacoste-Julien, and Nicolas Loizou. Dynamics of SGD with stochastic Polyak stepsizes: Truly adaptive variants and convergence to exact solution. May 2022.

Alasdair Paren, Leonard Berrada, Rudra P. K. Poudel, and M. Pawan Kumar. A stochastic bundle method for interpolation. *Journal of Machine Learning Research*, 23(15):1–57, 2022. URL http://jmlr.org/papers/v23/20-1248.html.

Adam Paszke, Sam Gross, Francisco Massa, Adam Lerer, James Bradbury, Gregory Chanan, Trevor Killeen, Zeming Lin, Natalia Gimelshein, Luca Antiga, Alban Desmaison, Andreas Kopf, Edward Yang, Zachary DeVito, Martin Raison, Alykhan Tejani, Sasank Chilamkurthy, Benoit Steiner, Lu Fang, Junjie Bai, and Soumith Chintala. Pytorch: An imperative style, high-performance deep learning library. In *Advances in Neural Information Processing Systems 32*, pp. 8024–8035. Curran Associates, Inc., 2019. URL http://papers.neurips.cc/paper/9015-pytorch-an-imperative-style-high-performance-deep-learning-library.pdf.

Boris T. Polyak. *Introduction to optimization*. Translations Series in Mathematics and Engineering. Optimization Software, Inc., Publications Division, New York, 1987. ISBN 0-911575-14-6. Translated from the Russian, With a foreword by Dimitri P. Bertsekas.

Mariana Prazeres and Adam M. Oberman. Stochastic gradient descent with Polyak's learning rate. *Journal of Scientific Computing*, 89(1):Paper No. 25, 16, 2021. ISSN 0885-7474. doi: 10.1007/s10915-021-01628-3.

L.M. Rivera-Muñoz, A.F. Giraldo-Forero, and J.D. Martinez-Vargas. Deep matrix factorization models for estimation of missing data in a low-cost sensor network to measure air quality. *Ecological Informatics*, 71: 101775, 2022. ISSN 1574-9541. doi: https://doi.org/10.1016/j.ecoinf.2022.101775. URL https://www.sciencedirect.com/science/article/pii/S1574954122002254.

Herbert Robbins and Sutton Monro. A stochastic approximation method. *Ann. Math. Statistics*, 22:400–407, 1951. ISSN 0003-4851. doi: 10.1214/aoms/1177729586.

R. Tyrrell Rockafellar and Roger J.-B. Wets. *Variational analysis*, volume 317 of *Grundlehren der mathematischen Wissenschaften [Fundamental Principles of Mathematical Sciences]*. Springer-Verlag, Berlin, 1998. ISBN 3-540-62772-3. doi: 10.1007/978-3-642-02431-3.

Nathan Srebro, Jason Rennie, and Tommi Jaakkola. Maximum-margin matrix factorization. In L. Saul, Y. Weiss, and L. Bottou (eds.), *Advances in Neural Information Processing Systems*, volume 17. MIT Press, 2004. URL https://proceedings.neurips.cc/paper/2004/file/e0688d13958a19e087e123148555e4b4-Paper.pdf.

Guodong Zhang, Chaoqi Wang, Bowen Xu, and Roger B. Grosse. Three mechanisms of weight decay regularization. In *7th International Conference on Learning Representations, ICLR 2019, New Orleans, LA, USA, May 6-9, 2019*. OpenReview.net, 2019. URL https://openreview.net/forum?id=B1lz-3Rct7.

Zhenxun Zhuang, Mingrui Liu, Ashok Cutkosky, and Francesco Orabona. Understanding AdamW through proximal methods and scale-freeness. *Transactions on Machine Learning Research*, 2022. URL https://openreview.net/forum?id=IKhEPWGdwK.

## Contents

## A Missing Proofs

### A.1 Proofs of model-based update formula

**Lemma 9.** *For $\lambda \geq 0$, let $\varphi(x) = \frac{\lambda}{2}\|x\|^2$ and let $g \in \partial f(x; s)$ and $C(s) \leq \inf_{z \in \mathbb{R}^n} f(z; s)$ hold for all $s \in \mathcal{S}$. For*

$$\psi_x(y; s) = f_x(y; s) + \varphi(y), \quad f_x(y; s) = \max\{f(x; s) + \langle g, y - x\rangle, C(s)\},$$

*consider the update*

$$x^{k+1} = \arg\min_{x \in \mathbb{R}^n} \psi_{x^k}(x; S_k) + \frac{1}{2\alpha_k}\|x - x^k\|^2. \tag{30}$$

*Denote $C_k := C(S_k)$ and let $g_k \in \partial f(x^k; S_k)$. Define*

$$\tau_k^+ := \begin{cases} 0 & \text{if } g_k = 0, \\ \min\left\{\alpha_k, \left(\frac{(1+\alpha_k\lambda)(f(x^k; S_k) - C_k) - \alpha_k\lambda\langle g_k, x^k\rangle}{\|g_k\|^2}\right)_+\right\} & \text{else.} \end{cases}$$

*Then, we have*

$$x^{k+1} = \frac{1}{1 + \alpha_k\lambda}x^k - \frac{\tau_k^+}{1 + \alpha_k\lambda}g_k = \frac{1}{1 + \alpha_k\lambda}\left(x^k - \tau_k^+ g_k\right) = \text{prox}_{\alpha_k\varphi}(x^k - \tau_k^+ g_k). \tag{31}$$

*Define $\tau_k := 0$ if $g_k = 0$ and $\tau_k := \min\left\{\alpha_k, \frac{(1+\alpha_k\lambda)(f(x^k; S_k) - C_k) - \alpha_k\lambda\langle g_k, x^k\rangle}{\|g_k\|^2}\right\}$ else. Then, it holds $\tau_k \leq \tau_k^+$ and*

$$\psi_{x^k}(x^{k+1}; S_k) = f(x^k; S_k) - \frac{\alpha_k\lambda}{1 + \alpha_k\lambda}\langle g_k, x^k\rangle - \frac{\tau_k}{1 + \alpha_k\lambda}\|g_k\|^2 + \varphi(x^{k+1}). \tag{32}$$

*Proof.* Note that $\max\{f(x^k; S_k) + \langle g_k, y - x^k\rangle, C_k\}$ is convex as a function of $y$. The update is therefore unique. First, if $g_k = 0$, then clearly $x^{k+1} = \text{prox}_{\alpha_k\varphi}(x^k) = \frac{1}{1+\alpha_k\lambda}x^k$ and (32) holds true. Now, let $g_k \neq 0$. The solution of (30) is either in $\{y|f(x^k; S_k) + \langle g_k, y - x^k\rangle < C_k\}$, or in $\{y|f(x^k; S_k) + \langle g_k, y - x^k\rangle > C_k\}$ or in $\{y|f(x^k; S_k) + \langle g_k, y - x^k\rangle = C_k\}$. We therefore solve three problems:

(P1) Solve

$$y^+ = \arg\min_y C_k + \frac{\lambda}{2}\|y\|^2 + \frac{1}{2\alpha_k}\|y - x^k\|^2.$$

Clearly, the solution is $y^+ = \frac{1}{1+\alpha_k\lambda}x^k$. This $y^+$ solves (30) if $f(x^k; S_k) + \langle g_k, y^+ - x^k\rangle < C_k$.

(P2) Solve

$$y^+ = \arg\min_y f(x^k; S_k) + \langle g_k, y - x^k\rangle + \frac{\lambda}{2}\|y\|^2 + \frac{1}{2\alpha_k}\|y - x^k\|^2.$$

The optimality condition is $0 = \alpha_k g_k + \alpha_k\lambda y^+ + y^+ - x^k$. Thus, the solution is $y^+ = \frac{1}{1+\alpha_k\lambda}(x^k - \alpha_k g_k)$. This $y^+$ solves (30) if $f(x^k; S_k) + \langle g_k, y^+ - x^k\rangle > C_k$.

(P3) Solve

$$y^+ = \arg\min_y \frac{\lambda}{2}\|y\|^2 + \frac{1}{2\alpha_k}\|y - x^k\|^2, \quad \text{s.t. } f(x^k; S_k) + \langle g_k, y - x^k\rangle = C_k.$$

The KKT conditions are given by

$$\alpha_k \lambda y + y - x^k + \mu g_k = 0,$$
$$f(x^k; S_k) + \langle g_k, y - x^k \rangle = C_k.$$

Taking the inner product of the first equation with $g_k$, we get

$$(1 + \alpha_k \lambda) \langle g_k, y \rangle - \langle g_k, x^k \rangle + \mu \|g_k\|^2 = 0.$$

From the second KKT condition we have $\langle g_k, y \rangle = C_k - f(x^k; S_k) + \langle g_k, x^k \rangle$, hence

$$(1 + \alpha_k \lambda)\big(C_k - f(x^k; S_k) + \langle g_k, x^k \rangle\big) - \langle g_k, x^k \rangle + \mu \|g_k\|^2 = 0.$$

Solving for $\mu$ gives $\mu = \frac{(1 + \alpha_k \lambda)(f(x^k; S_k) - C_k) - \alpha_k \lambda \langle g_k, x^k \rangle}{\|g_k\|^2}$. From the first KKT condition, we obtain

$$y^+ = \frac{1}{1 + \alpha_k \lambda}\big(x^k - \mu g_k\big) = \frac{1}{1 + \alpha_k \lambda}\Big(x^k - \frac{(1 + \alpha_k \lambda)(f(x^k; S_k) - C_k) - \alpha_k \lambda \langle g_k, x^k \rangle}{\|g_k\|^2} g_k\Big).$$

This $y^+$ solves (30) if neither (P1) nor (P2) provided a solution.

For all three cases, the solution takes the form $y^+ = \frac{1}{1 + \alpha_k \lambda}[x^k - t g_k] =: y(t)$. As $\|g_k\|^2 > 0$, the term $f(x^k; S_k) + \langle g_k, y(t) - x^k \rangle$ is strictly monotonically decreasing in $t$. We know $f(x^k; S_k) + \langle g_k, y(t) - x^k \rangle = C_k$ for $t = \mu$ (from (P3)). Hence, $f(x^k; S_k) + \langle g_k, y(t) - x^k \rangle < C_k$ ($> C_k$) if and only if $t > \mu$ ($t < \mu$).

We conclude:

- If $f(x^k; S_k) + \langle g_k, y(0) - x^k \rangle < C_k$, then the solution to (P1) is the solution to (30). This condition is equivalent to $\mu < 0$.

- If $f(x^k; S_k) + \langle g_k, y(\alpha_k) - x^k \rangle > C_k$, then the solution to (P2) is the solution to (30). This condition is equivalent to $\alpha_k < \mu$.

- If neither $0 > \mu$ nor $\alpha_k < \mu$ hold, i.e. if $\mu \in [0, \alpha_k]$, then the solution to (30) comes from (P3) and hence is given by $y(\mu)$.

Altogether, we get that $x^{k+1} = \frac{1}{1 + \alpha_k \lambda}[x^k - \tau_k^+ g_k]$ with $\tau_k^+ = \min\{\alpha_k, (\mu)_+\}$.
Now, we prove (32). Note that if $g_k \neq 0$, then $\tau_k = \min\{\alpha_k, \mu\}$ with $\mu$ defined as in (P3). In the case of (P1), we have $\psi_{x^k}(x^{k+1}; S_k) = C_k + \varphi(x^{k+1})$. Moreover, it holds $\mu < 0$ and as $\alpha_k > 0$ we have $\tau_k = \mu$. Plugging $\tau_k = \mu$ into the right hand-side of (32), we obtain $C_k + \varphi(x^{k+1})$.
In the case of (P2) or (P3), we have $C_k \leq f(x^k; S_k) + \langle g_k, x^{k+1} - x^k \rangle$. Due to $f(x^k; S_k) + \langle g_k, y(t) - x^k \rangle = f(x^k; S_k) - \frac{1}{1 + \alpha_k \lambda} \langle g_k, x^k \rangle + \frac{t}{1 + \alpha_k \lambda} \|g_k\|^2$, we obtain (32) as $x^{k+1} = y(\alpha_k)$ and $\mu > \alpha_k$ in the case of (P2) and $x^{k+1} = y(\mu)$ and $\mu \leq \alpha_k$ in the case of (P3). $\qquad \square$

**Lemma 10.** *Consider the model $f_x(y; s) := \max\{f(x; s) + \langle g, y - x \rangle, C(s)\}$ where $g \in \partial f(x; s)$ and $C(s) \leq \inf_{z \in \mathbb{R}^n} f(z; s)$ holds for all $s \in \mathcal{S}$. Then, update (5) is given as*

$$x^{k+1} = x^k - \gamma_k g_k, \quad \gamma_k = \begin{cases} 0 & \text{if } g_k = 0, \\ \min\Big\{\alpha_k, \frac{f(x^k; S^k) - C(S_k)}{\|g_k\|^2}\Big\} & \text{else.} \end{cases}$$

*where $g_k \in \partial f(x^k; S_k)$. Moreover, it holds*

$$f_{x^k}(x^{k+1}; S_k) = \max\{C(S_k), f(x^k; S_k) - \alpha_k \|g_k\|^2\}, \tag{33}$$

*and therefore $f_{x^k}(x^{k+1}; S_k) = f(x^k; S_k) - \gamma_k \|g_k\|^2$.*

*Proof.* We apply Lemma 9 with $\lambda = 0$. As $f(x^k; S_k) \geq C(S_k)$, we have that $\tau_k^+ = \tau_k = \gamma_k$. $\qquad \square$

## A.2 Proof of Theorem 7

From now on, denote with $\mathcal{F}_k$ the filtration that is generated by the history of all $S_j$ for $j = 0, \ldots, k-1$.

*Proof of Theorem 7.* In the proof, we will denote $g_k = \nabla f(x^k; S_k)$. We apply Lemma 6, (17) with $x = x^\star$. Due to Lemma 2 (ii) and convexity of $f(\cdot; s)$ it holds

$$\psi_{x^k}(x^\star; S_k) \le f(x^\star; S_k) + \varphi(x^\star).$$

Together with (18), we have

$$
\begin{aligned}
(1 + \alpha_k \lambda)\|x^{k+1} - x^\star\|^2 \le \|x^k - x^\star\|^2 &- \|x^{k+1} - x^k\|^2 + 2\alpha_k[\varphi(x^\star) - \varphi(x^{k+1})] \\
&+ 2\alpha_k\big[f(x^\star; S_k) - f(x^k; S_k) - \langle g_k, x^{k+1} - x^k\rangle\big].
\end{aligned}
\tag{34}
$$

Smoothness of $f$ yields

$$-f(x^k) \le -f(x^{k+1}) + \langle \nabla f(x^k), x^{k+1} - x^k\rangle + \tfrac{L}{2}\|x^{k+1} - x^k\|^2.$$

Consequently,

$$
\begin{aligned}
-\langle g_k, x^{k+1} - x^k\rangle &= f(x^k) - f(x^k) - \langle g_k, x^{k+1} - x^k\rangle \\
&\le f(x^k) - f(x^{k+1}) + \langle \nabla f(x^k) - g_k, x^{k+1} - x^k\rangle + \tfrac{L}{2}\|x^{k+1} - x^k\|^2 \\
&\le f(x^k) - f(x^{k+1}) + \frac{\theta\alpha_k}{2}\|\nabla f(x^k) - g_k\|^2 + \frac{1}{2\theta\alpha_k}\|x^{k+1} - x^k\|^2 + \tfrac{L}{2}\|x^{k+1} - x^k\|^2.
\end{aligned}
$$

for any $\theta > 0$, where we used Young's inequality in the last step. Plugging into (34) gives

$$
\begin{aligned}
(1 + \alpha_k \lambda)\|x^{k+1} - x^\star\|^2 \le \|x^k - x^\star\|^2 &+ \big[\alpha_k L + \tfrac{1}{\theta} - 1\big]\|x^{k+1} - x^k\|^2 + 2\alpha_k[\varphi(x^\star) - \varphi(x^{k+1})] \\
&+ 2\alpha_k\big[f(x^\star; S_k) - f(x^k; S_k) + f(x^k) - f(x^{k+1})\big] + \theta\alpha_k^2\|\nabla f(x^k) - g_k\|^2.
\end{aligned}
$$

Applying conditional expectation, we have $\mathbb{E}[f(x^\star; S_k)|\mathcal{F}_k] = f(x^\star)$ and

$$\mathbb{E}[-f(x^k; S_k) + f(x^k)|\mathcal{F}_k] = 0, \quad \mathbb{E}[\|\nabla f(x^k) - g_k\|^2|\mathcal{F}_k] \le \beta.$$

Moreover, by assumption, $\alpha_k L + \tfrac{1}{\theta} - 1 \le 0$. Altogether, applying total expectation yields

$$(1 + \alpha_k \lambda)\mathbb{E}\|x^{k+1} - x^\star\|^2 \le \mathbb{E}\|x^k - x^\star\|^2 + 2\alpha_k\mathbb{E}[\psi(x^\star) - \psi(x^{k+1})] + \theta\beta\alpha_k^2$$

which proves (21).

**Proof of a):** let $\alpha_k = \frac{1}{\lambda(k + k_0)}$. Denote $\Delta_k := \mathbb{E}\|x^k - x^\star\|^2$. Rearranging and summing (21), we have

$$\sum_{k=0}^{K-1} \mathbb{E}[\psi(x^{k+1}) - \psi(x^\star)] \le \sum_{k=0}^{K-1}\left[\frac{1}{2\alpha_k}\Delta_k - \frac{1 + \alpha_k\lambda}{2\alpha_k}\Delta_{k+1} + \frac{\theta\beta\alpha_k}{2}\right].$$

Plugging in $\alpha_k$, we have $\frac{1 + \alpha_k\lambda}{2\alpha_k} = \frac{\lambda(k + k_0)}{2} + \frac{\lambda}{2}$ and thus

$$\sum_{k=0}^{K-1} \mathbb{E}[\psi(x^{k+1}) - \psi(x^\star)] \le \sum_{k=0}^{K-1}\left[\frac{\lambda(k + k_0)}{2}\Delta_k - \frac{\lambda(k + 1 + k_0)}{2}\Delta_{k+1}\right] + \frac{\theta\beta}{2}\sum_{k=0}^{K-1}\frac{1}{\lambda(k + k_0)}.$$

Dividing by $K$ and using convexity of $\psi$[10], we have

$$\mathbb{E}\left[\psi\Big(\frac{1}{K}\sum_{k=0}^{K-1} x^{k+1}\Big) - \psi(x^\star)\right] \le \frac{\lambda k_0}{2K}\|x^0 - x^\star\|^2 + \frac{\theta\beta}{2\lambda K}\sum_{k=0}^{K-1}\frac{1}{k + k_0}.$$

---

[10]By assumption $f$ is convex and therefore $\psi$ is convex.

Finally, as $k_0 \geq 1$, we estimate $\sum_{k=0}^{K-1} \frac{1}{k+k_0} \leq \sum_{k=0}^{K-1} \frac{1}{k+1} \leq 1 + \ln K$ by Lemma 13 and obtain (22).

**Proof of b):** Similar to the proof above, we rearrange and sum (21) from $k = 0, \ldots, K-1$, and obtain

$$\sum_{k=0}^{K-1} \alpha_k \mathbb{E}[\psi(x^{k+1}) - \psi(x^\star)] \leq \frac{\|x^0 - x^\star\|^2}{2} + \frac{\theta\beta \sum_{k=0}^{K-1} \alpha_k^2}{2}.$$

We divide by $\sum_{k=0}^{K-1} \alpha_k$ and use convexity of $\psi$ in order to obtain the left-hand side of (23). Moreover, by Lemma 13 we have

$$\sum_{k=0}^{K-1} \alpha_k \geq 2\alpha(\sqrt{K+1} - 1), \quad \sum_{k=0}^{K-1} \alpha_k^2 \leq \alpha^2(1 + \ln K).$$

Plugging in the above estimates, gives

$$\mathbb{E}\Big[\psi\Big(\frac{1}{\sum_{k=0}^{K-1} \alpha_k} \sum_{k=0}^{K-1} \alpha_k x^{k+1}\Big) - \psi(x^\star)\Big] \leq \frac{\|x^0 - x^\star\|^2}{4\alpha(\sqrt{K+1} - 1)} + \frac{\theta\beta\alpha(1 + \ln K)}{4(\sqrt{K+1} - 1)}.$$

**Proof of c):** If $f$ is $\mu$–strongly–convex, then $\psi$ is $(\lambda + \mu)$–strongly convex and

$$\psi(x^\star) - \psi(x^{k+1}) \leq -\frac{\mu+\lambda}{2}\|x^{k+1} - x^\star\|^2.$$

From (21), with $\alpha_k = \alpha$, we get

$$(1 + \alpha(\mu + 2\lambda))\mathbb{E}\|x^{k+1} - x^\star\|^2 \leq \mathbb{E}\|x^k - x^\star\|^2 + \theta\beta\alpha^2.$$

Doing a recursion of the above from $k = 0, \ldots, K-1$ gives

$$\mathbb{E}\|x^K - x^\star\|^2 \leq (1 + \alpha(\mu + 2\lambda))^{-K}\|x^0 - x^\star\|^2 + \theta\beta\alpha^2 \sum_{k=1}^{K}(1 + \alpha(\mu + 2\lambda))^{-k}$$

Using the geometric series, $\sum_{k=1}^{K}(1 + \alpha(\mu + 2\lambda))^{-k} \leq \frac{1+\alpha(\mu+2\lambda)}{\alpha(\mu+2\lambda)} - 1 = \frac{1}{\alpha(\mu+2\lambda)}$, and thus

$$\mathbb{E}\|x^K - x^\star\|^2 \leq (1 + \alpha(\mu + 2\lambda))^{-K}\|x^0 - x^\star\|^2 + \frac{\theta\beta\alpha}{\mu + 2\lambda}.$$

$\square$

## A.3 Proof of Theorem 8

*Proof of Theorem 8.* In the proof, we will denote $g_k = \nabla f(x^k; S_k)$. By assumption $f$ is $\rho$-weakly convex and hence $\psi$ is $(\rho - \lambda)$-weakly convex if $\rho > \lambda$ and convex if $\rho \leq \lambda$. Hence, $\hat{x}^k := \text{prox}_{\eta\psi}(x^k)$ is well-defined for $\eta < 1/(\rho - \lambda)$ if $\rho > \lambda$ and for any $\eta > 0$ else. Note that $\hat{x}^k$ is $\mathcal{F}_k$–measurable. We apply Lemma 6, (17) with $x = \hat{x}^k$. Due to Lemma 2 (ii) it holds

$$\psi_{x^k}(\hat{x}^k; S_k) = f_{x^k}(\hat{x}^k; S_k) + \varphi(\hat{x}^k) \leq f(\hat{x}^k; S_k) + \frac{\rho_{S_k}}{2}\|\hat{x}^k - x^k\|^2 + \varphi(\hat{x}^k).$$

Together with (18), this gives

$$(1 + \alpha_k\lambda)\|x^{k+1} - \hat{x}^k\|^2 \leq (1 + \alpha_k\rho_{S_k})\|x^k - \hat{x}^k\|^2 - \|x^{k+1} - x^k\|^2$$
$$+ 2\alpha_k\Big(\varphi(\hat{x}^k) - \varphi(x^{k+1}) + f(\hat{x}^k; S_k) - f(x^k; S_k) - \langle g_k, x^{k+1} - x^k\rangle\Big)$$

Analogous to the proof of Theorem 7, due to Lipschitz smoothness, for all $\theta > 0$ we have

$$-f(x^k; S_k) - \langle g_k, x^{k+1} - x^k\rangle \leq -f(x^k; S_k) + f(x^k)$$
$$- f(x^{k+1}) + \frac{\theta\alpha_k}{2}\|\nabla f(x^k) - g_k\|^2 + \Big[\frac{1}{2\theta\alpha_k} + \frac{L}{2}\Big]\|x^{k+1} - x^k\|^2.$$

Plugging in gives

$$(1 + \alpha_k \lambda)\|x^{k+1} - \hat{x}^k\|^2 \leq (1 + \alpha_k \rho_{S_k})\|x^k - \hat{x}^k\|^2 + 2\alpha_k\Big(\varphi(\hat{x}^k) - \varphi(x^{k+1})\Big)$$
$$+ 2\alpha_k\big(f(\hat{x}^k; S_k) - f(x^k; S_k) + f(x^k) - f(x^{k+1}) + \tfrac{\theta\alpha_k}{2}\|\nabla f(x^k) - g_k\|^2\big)$$
$$+ \big[\tfrac{1}{\theta} + \alpha_k L - 1\big]\|x^{k+1} - x^k\|^2.$$

It holds $\mathbb{E}[f(\hat{x}^k; S_k) - f(x^k; S_k)|\mathcal{F}_k] = f(\hat{x}^k) - f(x^k)$ and $\mathbb{E}[\psi(\hat{x}^k)|\mathcal{F}_k] = \psi(\hat{x}^k)$. By Assumption 4, we have $\mathbb{E}[\|g_k - \nabla f(x^k)\|^2|\mathcal{F}_k] \leq \beta$. Altogether, taking conditional expectation yields

$$(1 + \alpha_k \lambda)\mathbb{E}[\|x^{k+1} - \hat{x}^k\|^2|\mathcal{F}_k] \leq (1 + \alpha_k \rho)\|x^k - \hat{x}^k\|^2 + 2\alpha_k \mathbb{E}\big[\psi(\hat{x}^k) - \psi(x^{k+1})|\mathcal{F}_k\big]$$
$$+ \alpha_k^2 \theta\beta + \big[\tfrac{1}{\theta} + \alpha_k L - 1\big]\mathbb{E}[\|x^{k+1} - x^k\|^2|\mathcal{F}_k].$$

Next, the definition of the proximal operator implies that almost surely

$$\psi(\hat{x}^k) + \tfrac{1}{2\eta}\|\hat{x}^k - x^k\|^2 \leq \psi(x^{k+1}) + \tfrac{1}{2\eta}\|x^{k+1} - x^k\|^2,$$

and hence

$$\mathbb{E}\big[\psi(\hat{x}^k) - \psi(x^{k+1})|\mathcal{F}_k\big] \leq \mathbb{E}\big[\tfrac{1}{2\eta}\|x^{k+1} - x^k\|^2 - \tfrac{1}{2\eta}\|\hat{x}^k - x^k\|^2|\mathcal{F}_k\big].$$

Altogether, we have

$$(1 + \alpha_k \lambda)\mathbb{E}[\|x^{k+1} - \hat{x}^k\|^2|\mathcal{F}_k] \leq (1 + \alpha_k(\rho - \eta^{-1}))\|x^k - \hat{x}^k\|^2$$
$$+ \alpha_k^2 \theta\beta + \big[\tfrac{1}{\theta} + \alpha_k L + \alpha_k \eta^{-1} - 1\big]\mathbb{E}[\|x^{k+1} - x^k\|^2|\mathcal{F}_k].$$

From assumption (25), we can drop the last term. Now, we aim for a recursion in $\text{env}_\psi^\eta$. Using that

$$\frac{1 + \alpha_k(\rho - \eta^{-1})}{1 + \alpha_k \lambda} = \frac{1 + \alpha_k \lambda - \alpha_k \lambda + \alpha_k(\rho - \eta^{-1})}{1 + \alpha_k \lambda} = 1 + \frac{\alpha_k(\rho - \eta^{-1} - \lambda)}{1 + \alpha_k \lambda} \leq 1 + \alpha_k(\rho - \eta^{-1} - \lambda),$$

we get

$$\mathbb{E}[\text{env}_\psi^\eta(x^{k+1})|\mathcal{F}_k] \leq \mathbb{E}[\psi(\hat{x}^k) + \frac{1}{2\eta}\|x^{k+1} - \hat{x}^k\|^2|\mathcal{F}_k]$$
$$\leq \underbrace{\psi(\hat{x}^k) + \frac{1}{2\eta}\|x^k - \hat{x}^k\|^2}_{=\text{env}_\psi^\eta(x^k)} + \frac{1}{2\eta}\big[\alpha_k(\rho - \eta^{-1} - \lambda)\big]\|x^k - \hat{x}^k\|^2 + \frac{\alpha_k^2}{2\eta}\theta\beta.$$

Now using $\|x^k - \hat{x}^k\| = \eta\|\nabla\text{env}_\psi^\eta(x^k)\|$ we conclude

$$\mathbb{E}[\text{env}_\psi^\eta(x^{k+1})|\mathcal{F}_k] \leq \text{env}_\psi^\eta(x^k) + \frac{\eta}{2}\big[\alpha_k(\rho - \eta^{-1} - \lambda)\big]\|\nabla\text{env}_\psi^\eta(x^k)\|^2 + \frac{\alpha_k^2}{2\eta}\theta\beta.$$

Due to (25), we have $\eta^{-1} + \lambda - \rho > 0$. Taking expectation and unfolding the recursion by summing over $k = 0, \ldots, K - 1$, we get

$$\sum_{k=0}^{K-1} \tfrac{\alpha_k}{2}(1 - \eta(\rho - \lambda))\mathbb{E}\|\nabla\text{env}_\psi^\eta(x^k)\|^2 \leq \text{env}_\psi^\eta(x^0) - \mathbb{E}[\text{env}_\psi^\eta(x^K)] + \sum_{k=0}^{K-1} \tfrac{\alpha_k^2}{2\eta}\theta\beta.$$

Now using that $\text{env}_\psi^\eta(x^K) \geq \inf \psi$ almost surely, we finally get

$$\sum_{k=0}^{K-1} \alpha_k \mathbb{E}\|\nabla\text{env}_\psi^\eta(x^k)\|^2 \leq \frac{2(\text{env}_\psi^\eta(x^0) - \inf \psi)}{1 - \eta(\rho - \lambda)} + \frac{\beta\theta}{\eta(1 - \eta(\rho - \lambda))} \sum_{k=0}^{K-1} \alpha_k^2, \tag{35}$$

which proves (26). Now choose $\alpha_k = \frac{\alpha}{\sqrt{k+1}}$ and divide (35) by $\sum_{k=0}^{K-1} \alpha_k$. Using Lemma 13 for $\sum_{k=0}^{K-1} \alpha_k$ and $\sum_{k=0}^{K-1} \alpha_k^2$, we have

$$\min_{k=0,\ldots,K-1} \mathbb{E}\|\nabla \mathrm{env}_\psi^\eta(x^k)\|^2 \leq \frac{\mathrm{env}_\psi^\eta(x^0) - \inf \psi}{\alpha(1 - \eta(\rho - \lambda))(\sqrt{K+1} - 1)} + \frac{\beta\theta}{2\eta(1 - \eta(\rho - \lambda))} \frac{\alpha(1 + \ln K)}{(\sqrt{K+1} - 1)}.$$

Choosing $\alpha_k = \frac{\alpha}{\sqrt{K}}$ instead, we can identify the left-hand-side of (35) as $\alpha\sqrt{K}\mathbb{E}\|\nabla \mathrm{env}_\psi^\eta(x_\sim^K)\|^2$. Dividing by $\alpha\sqrt{K}$ and using $\sum_{k=0}^{K-1} \alpha_k^2 = \alpha^2$, we obtain

$$\mathbb{E}\|\nabla \mathrm{env}_\psi^\eta(x_\sim^K)\|^2 \leq \frac{2(\mathrm{env}_\psi^\eta(x^0) - \inf \psi)}{\alpha(1 - \eta(\rho - \lambda))\sqrt{K}} + \frac{\beta\theta}{\eta(1 - \eta(\rho - \lambda))} \frac{\alpha}{\sqrt{K}}.$$

$\square$

## B   Auxiliary Lemmas

**Lemma 11** (Thm. 4.5 in (Drusvyatskiy & Paquette, 2019)). *Let $f$ be $L$-smooth and $\varphi$ be proper, closed, convex. For $\eta > 0$, define $\mathcal{G}_\eta(x) := \eta^{-1}\big(x - \mathrm{prox}_{\eta\varphi}(x - \eta\nabla f(x))\big)$. It holds*

$$\tfrac{1}{4}\|\nabla \mathrm{env}_\psi^{1/(2L)}(x)\| \leq \|\mathcal{G}_{1/L}(x)\| \leq \tfrac{3}{2}(1 + \tfrac{1}{\sqrt{2}})\|\nabla \mathrm{env}_\psi^{1/(2L)}(x)\| \quad \forall x \in \mathbb{R}^n.$$

**Lemma 12.** *Let $c \in \mathbb{R}, a, x^0 \in \mathbb{R}^n$ and $\beta > 0$ and let $\varphi : \mathbb{R}^n \to \mathbb{R} \cup \{\infty\}$ be proper, closed, convex. The solution to*

$$y^+ = \operatorname*{arg\,min}_{y \in \mathbb{R}^n} \quad \big(c + \langle a, y\rangle\big)_+ + \varphi(y) + \frac{1}{2\beta}\|y - x^0\|^2 \tag{36}$$

*is given by*

$$y^+ = \begin{cases} \mathrm{prox}_{\beta\varphi}(x^0 - \beta a), & \text{if } c + \langle a, \mathrm{prox}_{\beta\varphi}(x^0 - \beta a)\rangle > 0, \\ \mathrm{prox}_{\beta\varphi}(x^0), & \text{if } c + \langle a, \mathrm{prox}_{\beta\varphi}(x^0)\rangle < 0, \\ \mathrm{prox}_{\beta\varphi}(x^0 - \beta u a) & \text{else, for } u \in [0,1] \text{ such that } c + \langle a, \mathrm{prox}_{\beta\varphi}(x^0 - \beta u a)\rangle = 0. \end{cases} \tag{37}$$

**Remark 4.** *The first two conditions can not hold simultaneously due to uniqueness of the solution. If neither of the conditions of the first two cases are satisfied, we have to find the root of $u \mapsto c + \langle a, \mathrm{prox}_{\beta\varphi}(x^0 - \beta u a)\rangle$ for $u \in [0,1]$. Due to strong convexity of the objective in (36), we know that there exists a root and hence $y^+$ can be found efficiently with bisection.*

*Proof.* The objective of (36) is strongly convex and hence there exists a unique solution. Due to (Beck, 2017, Thm. 3.63), $y$ is the solution to (36) if and only if it satisfies first-order optimality, i.e.

$$\exists u \in \partial(\cdot)_+(c + \langle a, y\rangle) : \ 0 \in u a + \partial\varphi(y) + \frac{1}{\beta}(y - x^0). \tag{38}$$

Now, as $y = \mathrm{prox}_{\beta\varphi}(z) \iff 0 \in \partial\varphi(y) + \frac{1}{\beta}(y - z)$, it holds

$$(38) \iff \exists u \in \partial(\cdot)_+(c + \langle a, y\rangle) : \ 0 \in \partial\varphi(y) + \frac{1}{\beta}(y - (x^0 - \beta u a))$$

$$\iff \exists u \in \partial(\cdot)_+(c + \langle a, y\rangle) : \ y = \mathrm{prox}_{\beta\varphi}(x^0 - \beta u a).$$

We distinguish three cases:

1. Let $\bar{y} := \mathrm{prox}_{\beta\varphi}(x^0 - \beta a)$ and suppose that $c + \langle a, \bar{y}\rangle > 0$. Then $\partial(\cdot)_+(c + \langle a, \bar{y}\rangle) = \{1\}$ and hence $\bar{y}$ satisfies (38) with $u = 1$. Hence, $y^+ = \bar{y}$.

2. Let $\bar{y} := \text{prox}_{\beta\varphi}(x^0)$ and suppose that $c + \langle a, \bar{y} \rangle < 0$. Then $\partial(\cdot)_+(c + \langle a, \bar{y} \rangle) = \{0\}$ and hence $\bar{y}$ satisfies (38) with $u = 0$. Hence, $y^+ = \bar{y}$.

3. If neither the condition of the first nor of the second case of (37) are satisfied, then, as (38) is a necessary condition for the solution $y^+$, it must hold $c + \langle a, y^+ \rangle = 0$. Hence, there exists a $u \in \partial(\cdot)_+(c + \langle a, y^+ \rangle) = [0, 1]$ such that

$$c + \langle a, \text{prox}_{\beta\varphi}(x^0 - u\beta a) \rangle = 0.$$

$\square$

**Lemma 13.** *For any $K \geq 1$ it holds*

$$\sum_{k=0}^{K-1} \tfrac{1}{k+1} = 1 + \sum_{k=1}^{K-1} \tfrac{1}{k+1} \leq 1 + \int_0^{K-1} \tfrac{1}{s+1} ds = 1 + \ln K,$$

$$\sum_{k=0}^{K-1} \tfrac{1}{\sqrt{k+1}} \geq \int_0^K \tfrac{1}{\sqrt{s+1}} ds = 2\sqrt{K+1} - 2.$$

The following is a detailed version of Proposition 3. We refer to Section 4.3 for context.

**Proposition 14.** *Let Assumption 1 and Assumption 3 hold and assume that there is an open, convex set $U$ containing $\text{dom } \varphi$. Let $f(\cdot; s)$ be $\rho_s$–weakly convex for all $s \in \mathcal{S}$ and let $\rho = \mathbb{E}[\rho_S]$. Assume that there exists $G_s \in \mathbb{R}_+$ for all $s \in \mathcal{S}$, such that $\mathsf{G} := \sqrt{\mathbb{E}[G_S^2]} < \infty$ and*

$$\|g(x; s)\| \leq G_s \quad \forall g(x; s) \in \partial f(x; s), \ \forall x \in U. \tag{39}$$

*Then, $\psi_x(y; s)$ (given in (10)) satisfies the following:*

(B1) *It is possible to generate infinitely many i.i.d. realizations $S_1, S_2, \ldots$ from $\mathcal{S}$.*

(B2) *It holds $\mathbb{E}[f_x(x; S)] = f(x)$ and $\mathbb{E}[f_x(y; S)] \leq f(y) + \tfrac{\rho}{2}\|y - x\|^2$ for all $x, y \in \mathbb{R}^n$.*

(B3) *The mapping $\psi_x(\cdot; s) = f_x(\cdot; s) + \varphi(\cdot)$ is convex for all $x \in \mathbb{R}^n$ and all $s \in \mathcal{S}$.*

(B4) *For all $x, y \in U$ and $s \in \mathcal{S}$, it holds $f_x(x; s) - f_x(y; s) \leq G_s\|x - y\|$.*

*Proof.* The properties (B1)–(B4) are identical to (B1)–(B4) in (Davis & Drusvyatskiy, 2019, Assum. B), setting $r = \varphi$, $f_x(\cdot, \xi) = f_x(\cdot; s)$, $\eta = 0$, $\tau = \rho$, $\mathsf{L} = \mathsf{G}$, and $L(\xi) = G_s$. (B1) is identical to Assumption 1. (B2) holds due to Lemma 2, (ii), applying expectation and using the definition of $f$, i.e. $f(x) = \mathbb{E}[f(x; S)]$. (B3) holds due to Lemma 2, (i) and convexity of $\varphi$. For (B4), taking $g \in \partial f(x; s)$ and $x, y \in U$, we have

$$f_x(x; s) - f_x(y; s) \leq f(x; s) - f(x; s) - \langle g, y - x \rangle \leq \|g\|\|y - x\| \leq G_s\|x - y\|.$$

$\square$

## C   Model equivalence for `SGD`

In the unregularized case, the `SGD` update

$$x^{k+1} = x^k - \alpha_k g_k, \quad g_k \in \partial f(x^k; S_k),$$

can be seen as solving (5) with the model

$$f_x(y; s) = f(x; s) + \langle g, y - x \rangle, \quad g \in \partial f(x; s).$$

Now, consider again the regularized problem (2) with $\varphi(x) = \frac{\lambda}{2}\|x\|^2$ and update (8) .
On the one hand, the model $\psi_x(y; s) = f(x; s) + \varphi(x) + \langle g + \lambda x, y - x \rangle$ with $g \in \partial f(x; s)$ yields

$$x^{k+1} = x^k - \alpha_k(g_k + \lambda x^k) = (1 - \alpha_k \lambda)x^k - \alpha_k g_k. \tag{40}$$

On the other hand, the model $\psi_x(y; s) = f(x; s) + \langle g, y - x \rangle + \varphi(y)$ with $g \in \partial f(x; s)$ results in

$$x^{k+1} = \text{prox}_{\alpha_k \varphi}(x^k - \alpha_k g_k) = \frac{1}{1 + \alpha_k \lambda}\big[x^k - \alpha_k g_k\big] = (1 - \frac{\alpha_k}{1 + \alpha_k \lambda}\lambda)x^k - \frac{\alpha_k}{1 + \alpha_k \lambda}g_k. \tag{41}$$

Running (40) with step sizes $\alpha_k = \beta_k$ is equivalent to running (41) with step sizes $\frac{\alpha_k}{1+\alpha_k \lambda} = \beta_k \iff \alpha_k = \frac{\beta_k}{1-\beta_k \lambda}$. In this sense, standard `SGD` can be seen to be equivalent to proximal `SGD` for $\ell_2$–regularized problems.

# D  Additional information on numerical experiments

## D.1  Matrix Factorization

**Synthetic data generation:** We consider the experimental setting of the deep matrix factorization experiments in (Loizou et al., 2021), but with an additional regularization. We generate data in the following way: first sample $B \in \mathbb{R}^{q \times p}$ with uniform entries in the interval $[0, 1]$. Then choose $\upsilon \in \mathbb{R}$ (which will be our targeted inverse condition number) and compute $A = DB$ where $D$ is a diagonal matrix with entries from 1 to $\upsilon$ (equidistant on a logarithmic scale)[11]. In order to investigate the impact of regularization, we generate a noise matrix $E$ with uniform entries in $[-\varepsilon, \varepsilon]$ and set $\tilde{A} := A \odot (1+E)$. We then sample $y^{(i)} \sim N(0, I)$ and compute the targets $b^{(i)} = \tilde{A}y^{(i)}$. A validation set of identical size is created by the same mechanism, but computing its targets, denoted by $b^{(i)}_{\text{val}}$, via the original matrix $A$ instead of $\tilde{A}$. The validation set contains $N_{\text{val}} = N$ samples.

| Name | $p$ | $q$ | $N$ | $\upsilon$ | $r$ | $\varepsilon$ |
|------|-----|-----|-----|-----|-----|-----|
| `matrix-fac1` | 6 | 10 | 1000 | 1e-5 | 4 | 0 |
| `matrix-fac2` | 6 | 10 | 1000 | 1e-5 | 10 | 0.05 |

Table 1: Matrix factorization synthetic datasets.

**Model and general setup:** Problem (29) can be interpreted as a two-layer neural network without activation functions. We train the network using the squared distance of the model output and $b^{(i)}$ (averaged over a mini-batch) as the loss function. We run 50 epochs for different methods, step size schedules and values of $\lambda$. For each different instance, we do ten independent runs: each run has the identical training set and initialization of $W_1$ and $W_2$, but different shuffling of the training set and different samples $y^{(i)}$ for the validation set. In order to allow a fair comparison, all methods have identical train and validation sets across all runs. All metrics are averaged over the ten runs. We always use a batch size of 20.

## D.2  Plots for `matrix-fac2`

In this section, we plot additional results for Matrix Factorization, namely for the setting `matrix-fac2` of Table 1, see Fig. 10, Fig. 11, and Fig. 12. The results are qualitatively very similar to the setting `matrix-fac1`.

---

[11]Note that (Loizou et al., 2021) uses entries from 1 to $\upsilon$ on a *linear* scale which, in our experiments, did not result in large condition numbers even if $\upsilon$ is very small.

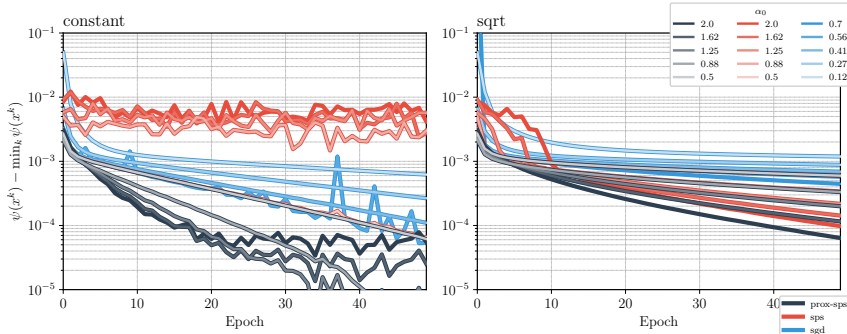

Figure 10: Objective function for the Matrix Factorization problem (29), with `constant` (left) and `sqrt` (right) step size schedule and several choices of initial values. Here $\min_k \psi(x^k)$ is the best objective function value found over all methods and all iterations.

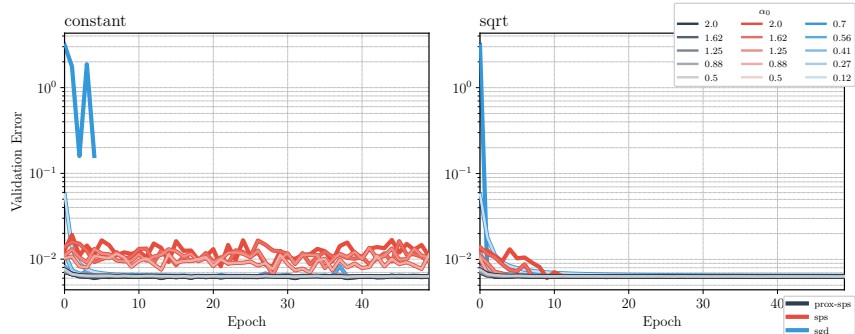

Figure 11: Validation error for the Matrix Factorization problem (29), with `constant` (left) and `sqrt` (right) step size schedule and several choices of initial values.

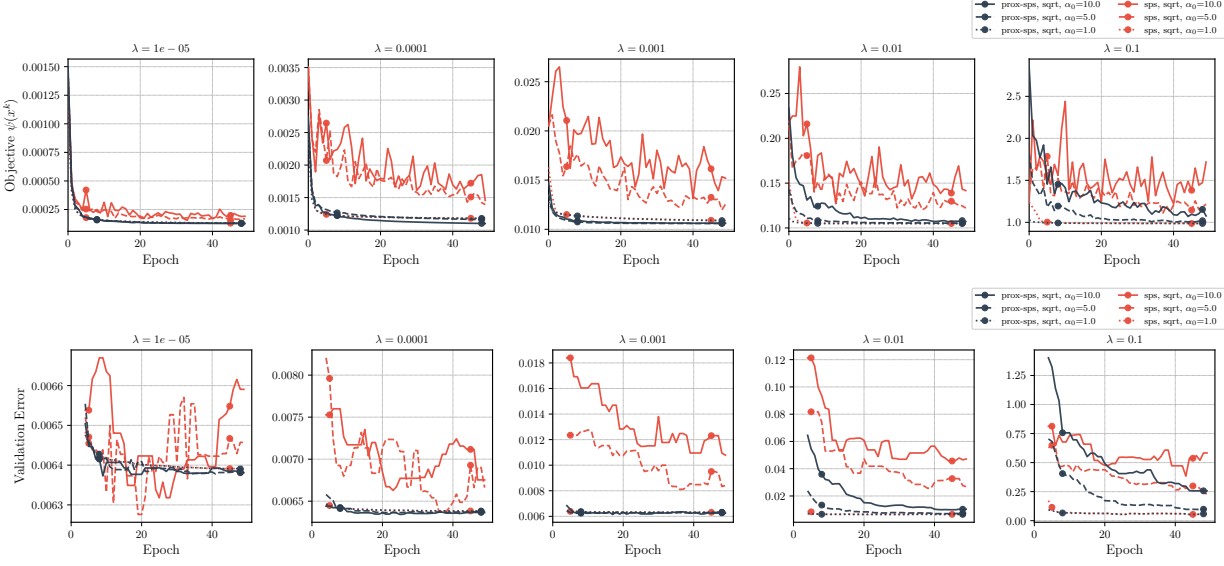

Figure 12: Objective function value and validation error over the course of optimization. For the validation error, we plot a rolling median over five epochs in order to avoid clutter.

### D.3 Matrix completion experiment

Consider an unknown matrix of interest $W \in \mathbb{R}^{d_1 \times d_2}$. Factorizing $W \approx U^\top V$ with $U \in \mathbb{R}^{r \times d_1}$, $V \in \mathbb{R}^{r \times d_2}$, we can estimate the entries of matrix $W$ as

$$\hat{W}_{ij} = u_i^\top v_j + b_i^U + b_j^V, \quad i \in [d_1], \ j \in [d_2], \tag{42}$$

where $u_i$ is the $i$-th column of $U$ and $v_j$ is the $j$-th column of $V$, and $b^U \in \mathbb{R}^{d_1}$, $b^V \in \mathbb{R}^{d_2}$ are bias terms (Rivera-Muñoz et al., 2022).

We can interpret this as an empirical risk minimization problem as follows: let $\mathcal{T}$ be the set of indices $(i, j)$ where $W_{ij}$ is known. With $\hat{W}_{ij}$ as in (42) for trainable parameters $(U, V, b^U, b^V)$, the (regularized) problem is then given as

$$\min_{U,V,b^U,b^V} \frac{1}{|\mathcal{T}|} \sum_{(i,j) \in \mathcal{T}} (W_{ij} - \hat{W}_{ij})^2 + \frac{\lambda}{2} \|(U, V, b^U, b^V)\|^2.$$

We use a dataset containing air quality measurements of a sensor network over one month. This dataset has been studied in Rivera-Muñoz et al. (2022).[12] The dataset contains measurements from 130 sensors over 720 timestamps, hence $d_1 = 130$, $d_2 = 720$. In total, there are 56158 nonzero measurements (the rest was missing data or removed due to being an outlier). We split the nonzero measurements into a training set of size $|\mathcal{T}| = 44926 \approx 0.8 \cdot 56158$ and the rest as a validation set. We standardize training and validation set using mean and variance of the training set. We set $r = 24$ and use batch size 128. The validation error is defined as the root mean squared error on the elements of the validation set (which is not used for training).

**Discussion**: The results are plotted in Fig. 13 and Fig. 14a. For all methods, we use a constant step size $\alpha_k$. ProxSPS achieves the smallest error on the validation set for the two smaller values of $\lambda$. For the largest $\lambda$, ProxSPS, SPS and SGD are almost identical for $\alpha_0 = 5$, but SGD with $\alpha_0 = 1$ is the best method. However, over all tested values of $\lambda$, Fig. 14a shows that ProxSPS obtains the smallest error. Again, from the lower plot in Fig. 13 we can observe that ProxSPS produces iterates with smaller norm.

### D.4 Imagenet32 experiment

Imagenet32 contains 1,28 million training and 50,000 test images of size $32 \times 32$, from 1,000 classes. We train the same ResNet110 as described in Section 5.3 with two differences: we exchange the output dimension of the final layer to 1,000 and activate batch norm. We use batch size 512. For this experiment we only run one repetition.

Similar to the setup in Section 5.3, we run all methods for three different values of $\lambda$. For AdamW, we use a constant learning rate 0.001, for SGD, SPS, and ProxSPS we use the sqrt-schedule and $\alpha_0 = 1$. The validation accuracy and model norm are plotted in Fig. 15: we can observe that all methods perform similarly well in terms of accuracy. However, AdamW is more sensitive with respect to the choice of $\lambda$ and the norm of its iterates differs significantly from the other methods. Further, using an adaptive step size is advantageous: from Fig. 16, we see that the adaptive step size is active in the initial iterations, which leads to a faster learning of (Prox)SPS in the initial epochs compared to SGD.

### D.5 Interpolation constant

We illustrate how the interpolation constant $\sigma^2$ behaves if it would be computed for the regularized loss $\ell_i(x) = f_i(x) + \frac{\lambda}{2}\|x\|^2$ (cf. also Section 4.2). We do a simple ridge regression experiment. Let $A \in \mathbb{R}^{N \times n}$ be a matrix with row vectors $a_i \in \mathbb{R}^n$, $i \in [N]$. We set $N = 80$, $n = 100$ and generate $\hat{x} \in \mathbb{R}^n$ with entries drawn uniformly from $[0, 1]$. We compute $b = A\hat{x}$. In this case, we have $f_i(x) = \frac{1}{2}(a_i^\top x - b_i)^2$ and $f(x) = \frac{1}{N} \sum_{i=1}^N f_i(x)$.

---

[12]The dataset can be downloaded from `https://github.com/andresgiraldo3312/DMF/blob/main/DatosEliminados/Ventana_Eli_mes1.csv`.

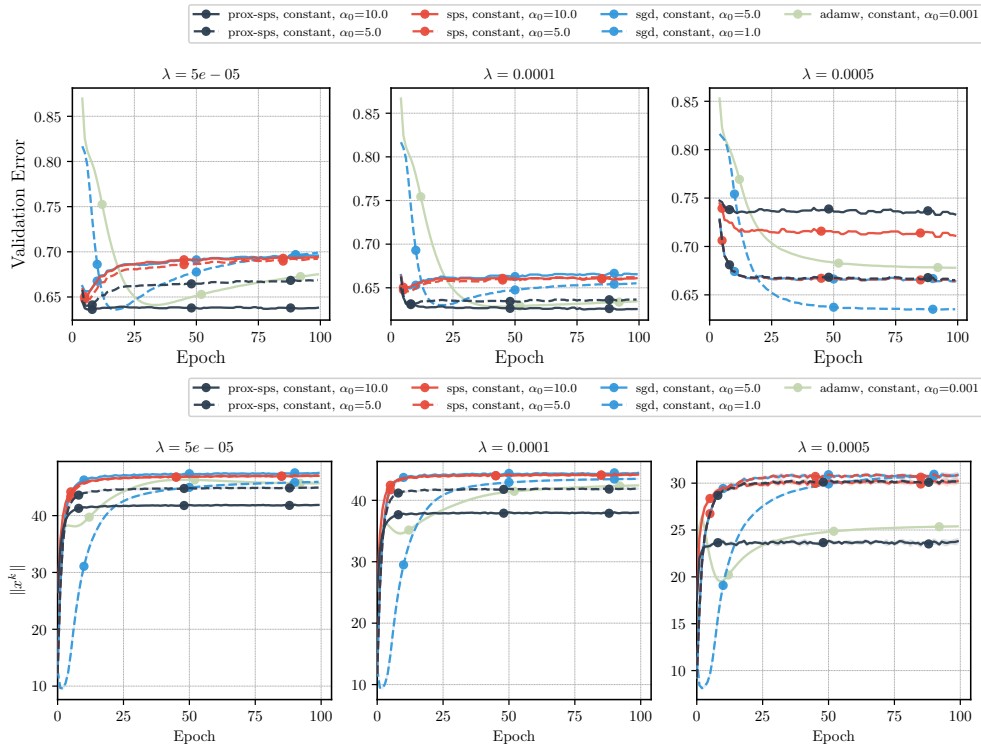

Figure 13: `Matrix Completion`: Validation error (top) and model norm (top) for three values of the regularization parameter $\lambda$. Validation error is plotted as five-epoch running median. Shaded area is two standard deviations over ten independent runs.

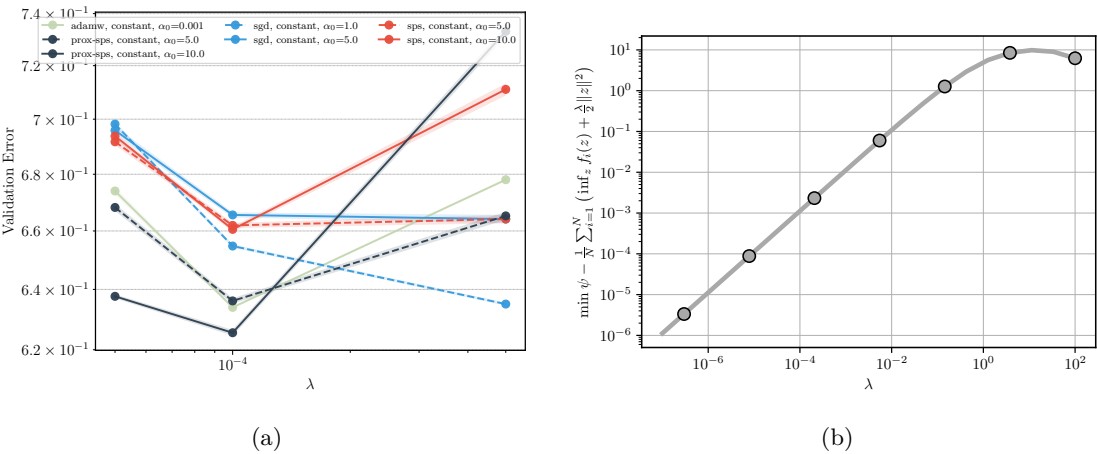

Figure 14: (a) `Matrix Completion`: Validation error as a function of the regularization parameter $\lambda$. Shaded area is one standard deviation (computed over ten independent runs). For all values, we take the median over epochs $[90, 100]$. (b) Interpolation constant for a ridge regression problem for varying regularization parameter $\lambda$. See Appendix D.5 for details.

If one would apply the theory of $\texttt{SPS}_{\max}$ for the regularized loss functions $\ell_i$ with estimates $\underline{\ell}_i = 0$, the constant $\sigma^2 = \left( \min_{x \in \mathbb{R}^n} f(x) + \varphi(x) \right) - \frac{1}{N} \sum_{i=1}^{N} \inf_z \ell_i(z)$ determines the size of the constant term in the convergence results of (Loizou et al., 2021; Orvieto et al., 2022). We compute $\min_{x \in \mathbb{R}^n} f(x) + \varphi(x)$ by solving the ridge regression problem. Further, the minimizer of $\ell_i$ is given by $(a_i a_i^\top + \lambda \mathbf{Id})^{-1} a_i b_i$. We plot $\sigma^2$ for varying $\lambda$ in Fig. 14b to verify that $\sigma^2$ grows significantly if $\lambda$ becomes large (even if the loss could

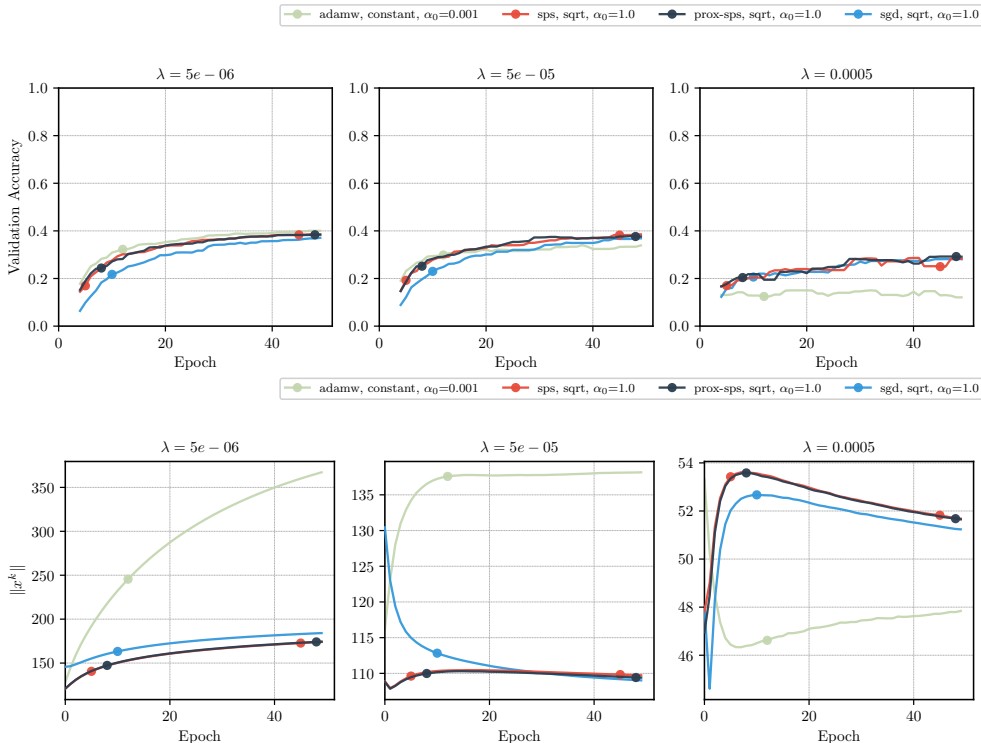

Figure 15: `ResNet110` for `Imagenet32`: Validation accuracy as five-epoch running median (top) and model norm (bottom) for three values of $\lambda$.

be interpolated perfectly, i.e. $\inf_x f(x) = 0$). We point out that the constant $\sigma^2$ does not appear in our convergence results Theorem 7 and Theorem 8.

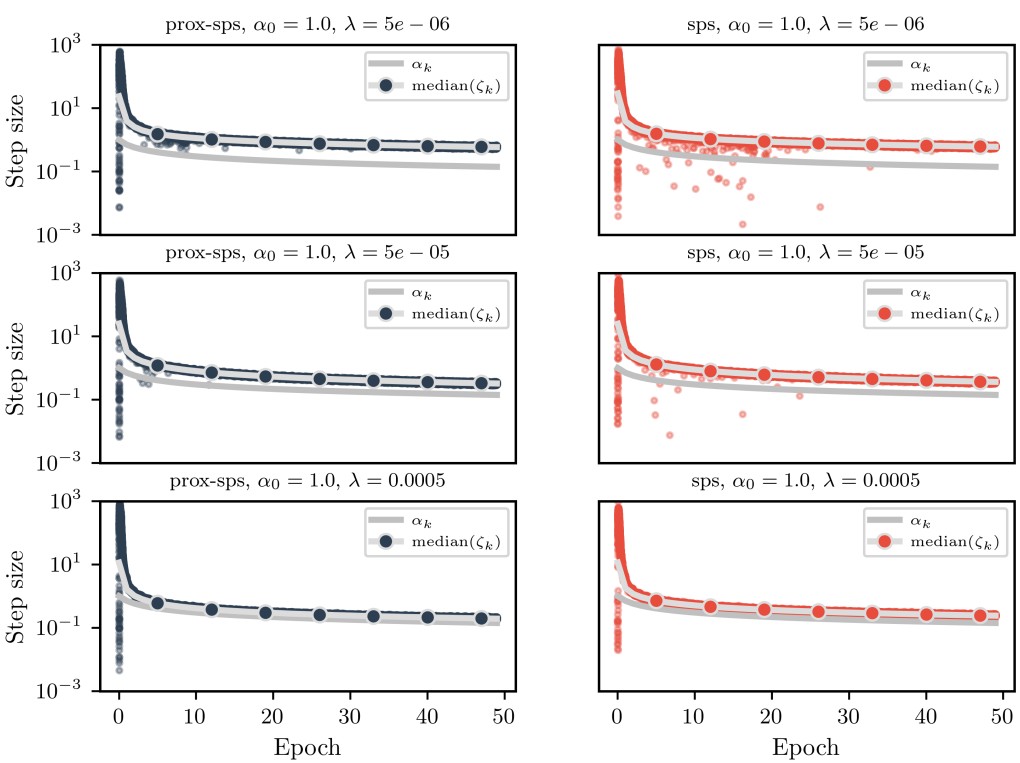

Figure 16: `ResNet110` for `Imagenet32`: Adaptive step sizes for `SPS` and `ProxSPS`. See definition of $\zeta_k$ in Section 5.1.

