# OpenReview forum: "A Stochastic Proximal Polyak Step Size"
_TMLR — Accepted by TMLR_

### Review · Reviewer_J2D7 · 2023-01-30

**Summary Of Contributions:**

This paper proposed a stochastic proximal update with Polyak step size for regularized empirical risk minimization problem with a model-based viewpoint. Different from the previous methods that also construct the model for the regularizer, the authors proposed to construct the model only for the risk and then perform the update with the proximal point method. The authors demonstrated that, with a decaying step-size parameter, the proposed method can exactly converge to the stationary point, which is in contrast to the previous methods.

**Audience:**

Yes

**Broader Impact Concerns:**

Not applicable.

**Claims And Evidence:**

Yes

**Requested Changes:**

Please provide a much more detailed comparison with the existing work, include but not limited to:

* For Loizou et al, 2021 and Orvieto et al., 2022, if they use a decaying step-size, can they obtain exact convergence?
* How does $\lambda$ affect the convergence results with a decaying step-size for the original method and the proposed method in theory?

I'm also interested in the reason that the proposed method still converges well with constant step-size but the original method fails. Can the authors provide some theoretical justification on that?

**Strengths And Weaknesses:**

### Strengths
The idea overall is clear and easy to follow.

The authors provide theoretical analysis and empirical justification.

### Weaknesses
The proposed method is not substantially novel. In fact it only replaces the step-size in original proximal point method with the Polyak step-size.

The comparison with the existing methods is a little bit confusing.

The experiment part is not aligned with the theoretical analysis well. Specifically, the authors focused on the decaying step-size in the main text, but in the experiments the authors also showed the results with constant step-size and claimed the original version failed in this case but the proposed method can work. Also, the authors claim that the proposed method work better with larger $\lambda$, but the theoretical analysis does not show that.

---

### Review · Reviewer_1wix · 2023-03-05

**Summary Of Contributions:**

This paper studies a variant of SGD for both smooth and non-smooth objective and also for the objectives with the composite structure. The only difference with the classical SGD is the choice of the step size. In this paper it is used as a combination of the Polyak step, adjusted for stochastic setting, and truncation idea.

Under either bounded subgradients or smoothness assumption of the main function $f$ and weakly-convexity of $f$ some convergence results are obtained.

**Audience:**

No

**Broader Impact Concerns:**

 -

**Claims And Evidence:**

No

**Requested Changes:**

Some comments:

1. I don't think it is a good idea to use the same letter $f$ to denote both functions. Sometimes it is not clear to which $f$ is referred in the text.

2. First $g$ denotes a (sub)-gradient, then it denotes a convex function, then again it is a subgradient.

3. Assumption 2: Does it mean that you assume that $\inf_x f(x; s)$ is finite? If this is the case, then it should be stated somehow in this form. Then $C(s)$ is just a notation.

4. page 5, first paragraph: I didn't understand: which $c$ is better in the end and why?
5. Why do we need to spend one page on special (and obvious) case for $\ell_2$ norm?
I would say just writing one equation should suffice.

6. It is a bit strange to read: in the intro $f$ is assumed to be convex, then it becomes weakly-convex.

7. I don't know what authors had in mind, but I don't see how the paper with the Proposition 3 in that form can be accepted.
8. Assumption 4 is quite strong, there are more flexible assumptions for this.
9. Corollary 4. It says that in order to run the Algorithm, we need to know $\Delta$, which is uncomputable in our setting: neither $\mathrm{env}_{\psi(x^0)}$, nor $\min_x \psi(x)$ are available. If we take it too large, then the complexity becomes vacuous.
10. Theorem 8. Haven't we already assumed that $\inf \psi > -\infty$?

**Strengths And Weaknesses:**


## Theory
From a theoretical point of view, all obtained results are a simple application of the Davis & Drusvyatskiy 2019 paper. That is, there is not much insight behind the analysis and the paper doesn't bring any new ideas on this front. Such results are already obtained for SGD and other variants, often with less restrictive assumptions. The theoretical guarantees of the algorithm are not better and actually even worse if we take into account that the algorithm is not implementable (see the comment on Corollary 4 below).

The minor addition like adding the prox function is also straightforward.

### The truncation idea
The authors suggest to choose step size $\gamma_k$ as
$$ \gamma_k = \min \left(\alpha_k, {\color{blue}\mathrm{some\ expression}} \right)$$

and then still require $\alpha_k$ to be of the order $1/\sqrt{k}$ or $1/k$, depending on the properties of $f$. I don't understand why we need this. The ${\color{blue}\mathrm{some\ expression}}$ is well separated from zero (otherwise the problem is very special), so eventually $\min$ will disappear and we end up just with the step $\alpha_k$, exactly as in the classic SGD. Isn't it obvious that in the beginning of SGD we can use any step sizes whatsoever and then switch to a proper "convergent regime"?




## Motivation
Roughly speaking: why should we care? There are so many variations of SGD. Neither theory nor a basic intuition suggests to me that this should be an advantageous algorithm. Since it requires the same steps in the end, relying on numerics as a possible motivation doesn't make sense to me either.

---

### Review · Reviewer_24qn · 2023-03-09

**Summary Of Contributions:**

This work extends the stochastic Polyak step size SPS with a proximal variant called ProxSPS, which can handle regularization terms efficiently.  The results show that ProxSPS is easier to tune and more stable in the presence of regularization than SPS for datasets like CIFAR10 with large networks like ResNet110.

Compared to the stochastic Polyak step size SPS, ProxSPS only requires estimating a lower bound for the loss function and not for the composite function (that includes the L2 regularization term). Further, ProxSPS has closed-form updates for squared L2-regularization and provides theoretical guarantees for proper, closed, and convex regularization function.

**Audience:**

Yes

**Broader Impact Concerns:**

It is not clear how much tuning is required for this method to achieve good performance on some problem setups, which can lead to large unnecessary usage of compute like running large-scale GPU resources which can negatively contribute to the environment.

**Claims And Evidence:**

Yes

**Requested Changes:**

Compare between ProxSPS, SPS, SGD and Adam on the following two useful tasks:

- ImageNet classification.

- Fine-tuning a pre-trained large language model like BERT for classifing a text-classification problem like CLINC150.

The reason for these experiments is to see how ProxSPS compares to other methods in large scale settings.


**Strengths And Weaknesses:**

Strengths
-----------
This work is well written and the authors address a task that is important for the community which could have impact on
the field of efficiently training deep learning methods.

I appreciate the simplicity of the method and the fact that it is easy to implement.

The authors also provide a nice discussion of the theoretical properties of the method.

ProxSPS seems to perform better than SPS when larger networks are used, which is a nice result as it shows that the method
is not limited to small networks and there is potential that it can be used to train large networks like LLMs.

The authors performed multiple runs in their experiments and report the mean and standard deviation of the results
which is important for understanding the performance of the method.

The experiments are exhaustive and the authors provide a nice discussion of the results and how they compare to other methods in standard datasets. It is especially interesting to see ProxSPS works well with larger networks like ResNet110 vs. SPS, suggesting that it might be able to work well for larger networks like LLMs.


Weaknesses
--------------
The authors should test this method on larger datasets like ImageNet and LLM type of datasets in order to identify the limitations of the method in large scale settings and see how it compares against standard methods like AdamW.

---

### Author Response · Authors · 2023-04-04
**Summary of our Revision**

Dear Reviewers,

we thank you for your constructive feedback. We have just uploaded our revised article - the changes compared to the initial submission are in red. Here is a short summary:

* Based on the comments of Reviewer 24qn, we added an experiment for the Imagenet32 dataset. The results are in section D.4.
* We added a further experiment for a matrix completion task on a real-life dataset. The results are in section D.3. We defer to both of these new experiments in the main text.
* Based on the comments of Reviewer J2D7, we added more explanations in the section "4.3.3. Comparison to existing theory". We hope that this clarifies even more how our results differ from the existing ones - in particular we explain why the dependence on the interpolation constant (as in Loizou et al., 2021) is suboptimal in the regularized setting. We provide a simple numerical example to validate this observation in section D.5.
* Based on the comments of Reviewer 1wix, we rephrased and further explained Proposition 3. A detailed version is now given in the Appendix where we precisely formulate the properties of (Davis & Drusvyatskiy, 2019, Assum. B).

We believe that the review period led to improvements in clarity and experimental validation of the paper. We hope that all of your questions have been answered and are happy to discuss further if needed.